# Open syntaxin overcomes exocytosis defects of diverse mutants in *C. elegans*

Chi-Wei Tien [1,2,13], Bin Yu [3,13], Mengjia Huang [1,2], Karolina P. Stepien [4,5,6], Kyoko Sugita[7], Xiaoyu Xie[1,8], Liping Han[8,9], Philippe P. Monnier[2,7,10], Mei Zhen [2,11,12], Josep Rizo [4,5,6 ✉], Shangbang Gao [3 ✉] & Shuzo Sugita [1,2 ✉]

Assembly of SNARE complexes that mediate neurotransmitter release requires opening of a 'closed' conformation of UNC-64/syntaxin. Rescue of *unc-13/Munc13* mutant phenotypes by overexpressed open UNC-64/syntaxin suggested a specific function of UNC-13/Munc13 in opening UNC-64/ syntaxin. Here, we revisit the effects of open *unc-64/*syntaxin by generating knockin (KI) worms. The KI animals exhibit enhanced spontaneous and evoked exocytosis compared to WT animals. Unexpectedly, the open syntaxin KI partially suppresses exocytosis defects of various mutants, including *snt-1/*synaptotagmin, *unc-2/*P/Q/N-type $Ca^{2+}$ channel alpha-subunit and *unc-31/*CAPS, in addition to *unc-13/*Munc13 and *unc-10/*RIM, and enhanced exocytosis in *tom-1/*Tomosyn mutants. However, open syntaxin aggravates the defects of *unc-18/*Munc18 mutants. Correspondingly, open syntaxin partially bypasses the requirement of Munc13 but not Munc18 for liposome fusion. Our results show that facilitating opening of syntaxin enhances exocytosis in a wide range of genetic backgrounds, and may provide a general means to enhance synaptic transmission in normal and disease states.

[1] Division of Fundamental Neurobiology, Krembil Brain Institute, University Health Network, Toronto, Ontario, Canada M5T 2S8. [2] Faculty of Medicine, Department of Physiology, University of Toronto, Toronto, Ontario, Canada M5S 1A8. [3] Key Laboratory of Molecular Biophysics of the Ministry of Education, College of Life Science and Technology, Huazhong University of Science and Technology, Wuhan 430074, China. [4] Department of Biophysics, University of Texas Southwestern Medical Center, Dallas, Texas, USA. [5] Department of Biochemistry, University of Texas Southwestern Medical Center, Dallas, Texas, USA. [6] Department of Pharmacology, University of Texas Southwestern Medical Center, Dallas, Texas, USA. [7] Division of Genetics and Development, Krembil Brain Institute, University Health Network, Toronto, Ontario, Canada M5T 2S8. [8] Department of Anesthesia, Dalian Medical University, Dalian, Liaoning, China. [9] Department of Anesthesiology, Dalian Municipal Friendship Hospital, Dalian Medical University, Dalian, Liaoning, China. [10] Department of Ophthalmology and Vision Sciences, University of Toronto, Toronto, Ontario, Canada M5S 1A8. [11] Lunenfeld-Tanenbaum Research Institute, Mount Sinai Hospital, Toronto, Ontario, Canada M5G 1X5. [12] Faculty of Medicine, Department of Molecular Genetics, University of Toronto, Toronto, Ontario, Canada M5S 1A8. [13]These authors contributed equally: Chi-Wei Tien, Bin Yu. ✉email: Jose.rizo-rey@utsouthwestern.edu; sgao@hust.edu.cn; Shuzo.Sugita@uhnresearch.ca

Synaptic vesicle exocytosis provides the chemical basis for neuronal communication, enabling the amazing diversity of functions of the brain. The SNARE (soluble NSF-attachment receptor) proteins syntaxin, synaptobrevin/VAMP, and SNAP-25 play a central role in exocytosis by forming a tight SNARE complex[1] that consists of a four-helix bundle[2,3]. This four-helix bundle is believed to be partially formed before $Ca^{2+}$ influx[4]. $Ca^{2+}$, acting through synaptotagmin-1[5,6], triggers full zippering of the SNARE complex, which ultimately results in fusion of the vesicle with the plasma membrane.

A biochemical step preceding the full fusion is often referred to as "priming." It involves formation of the fusion-competent state of synaptic vesicles just prior to $Ca^{2+}$ entry. The priming process is tightly controlled by multiple proteins, including Munc18-1/UNC-18[7–9], Munc13-1/2/UNC-13[10–12], Tomosyn[13–15], and RIM/UNC-10[16,17]. However, the precise mechanisms through which these proteins mediate priming is still unclear.

Syntaxin/UNC-64 is a key SNARE protein that is crucial for exocytosis. It exists in an open and a closed conformation, the latter forming a tight complex with Munc18-1/UNC-18[18]. The binding of closed syntaxin and Munc18-1/UNC-18 has been shown to be critical for the level and plasmalemmal localization of syntaxin[7,19–22]. However, when held closed by Munc18-1/UNC-18, syntaxin is unable to interact with synaptobrevin and SNAP-25 to form SNARE complexes[23,24]. Munc13-1/UNC-13 was postulated to open syntaxin and consequently promote SNARE complex assembly and subsequent exocytosis at axon terminals. This important hypothesis originated from studies in which multi-copy overexpression of a constitutively open form of syntaxin/UNC-64 in Caenorhabditis elegans partially restored motility and strongly rescued synaptic vesicle exocytosis defects of unc-13 null worms[25]. The rescue was suggested to be due to a specific genetic interaction between unc-13 and syntaxin, as expression of the open syntaxin in unc-64-null (js21) mutant[26] did not enhance exocytosis compared with the wild-type (WT) syntaxin and did not rescue the reduced exocytosis caused by reduced $Ca^{2+}$ influx in a severe loss-of-function mutant (e55) of UNC-2 $Ca^{2+}$ channels[25]. This model also received support from diverse biochemical experiments, including the demonstration that the MUN domain of Munc13-1 enhances SNARE complex assembly starting from the Munc18-1-closed syntaxin-1 complex[27–29].

However, multicopy expression of open syntaxin was also found to rescue the exocytosis defects in unc-10/RIM mutants[17] and dense core vesicle-docking defects in unc-31/CAPS mutants[30], calling into question the specificity of the genetic interaction between open syntaxin and unc-13, as well as others. Strong overexpression of syntaxin may not inform on the full physiological state or functions of the endogenous gene products. Indeed, the extent of open syntaxin-mediated restoration of evoked neurotransmitter release in unc-13-null mutant by these transgenes was variable in C. elegans[15,25].

Moreover, the reported phenotype of multicopy open syntaxin in C. elegans was quite different from that of open syntaxin-1B knock-in (KI) mice[31,32]. In particular, open syntaxin-1B led to enhanced $Ca^{2+}$ sensitivity and fusion competency, as well as an increase in spontaneous release in hippocampal neurons[31], which was not observed in open syntaxin worms[25]. In measurements performed at the Calyx of Held, the open syntaxin-1B mutation enhanced the speed of evoked release and accelerated fusion pore expansion[32]. These phenotypes were postulated to arise from an increase in the number of SNARE complexes formed on vesicles, as a consequence of the facilitation of syntaxin-1B opening. However, use-dependent depression of excitatory postsynaptic currents (EPSCs) was increased in hippocampal synapses, suggesting a reduction in readily releasable pool (RRP) size in the open syntaxin-1B mutants due to reduced interactions between open syntaxin-1B and Munc18-1, and a corresponding decrease in the levels of both proteins[31]. Furthermore, open syntaxin-1 was reported to fail or yield very limited rescue of phenotypes observed in Munc13-1 knockout (KO) or Munc13-1/2 double KO (DKO) mice[31,33].

In this study, we re-evaluated the effects of the open form of syntaxin by generating the first in vivo KI model of the open syntaxin mutant (L166A/E167A, often called L165E/E166A (LE0 mutant) in C. elegans. We then introduced the LE open mutation into various exocytosis mutant backgrounds to systematically examine the genetic interactions between open syntaxin and other regulators of exocytosis. The phenotype of open syntaxin and its genetic interactions with other synaptic transmission mutants only shared limited similarity with the ones previously described using multicopy transgenes[25]. Our data show that the open syntaxin KI enhances neurotransmitter release and can partially rescue the phenotypes of a wide variety of exocytosis mutants except for Munc18/UNC-18. These results support the proposal that increasing the number of assembled SNARE complexes leads to enhanced neurotransmitter release and suggest that controlling the opening of syntaxin provides a general avenue to modulate synaptic transmission, which may help to develop therapeutic strategies for neurological diseases with impaired synaptic function.

## Results

**The open syntaxin KI mutant enhances spontaneous exocytosis compared to WT C. elegans.** The L165A/E166A mutation in mammalian syntaxin-1A is known to result in a constitutively open form of syntaxin[18]. Previous studies demonstrated that multicopy expression of UNC-64, the C. elegans syntaxin, harboring the analogous L166A and E167A mutations led to slower-moving animals than the WT without affecting the level of exocytosis[25]. To determine the effects of a KI open syntaxin mutation on exocytosis, we generated unc-64(sks4) mutants where we introduced an L166A and E167A mutation in the unc-64 endogenous locus by CRISPR/Cas9-directed homologous recombination (Supplementary Fig. S1)[34]. The genomic DNA sequence of open syntaxin/unc-64 is shown in comparison with WT in Supplementary Fig. S2. To quantify the expression level of open syntaxin (=the UNC-64(sks4) mutant) in comparison with N2 WT syntaxin (=WT UNC-64), we performed several ($n = 7$) western blot analyses (Supplementary Fig. S1c). We used β-tubulin signals as loading controls. The quantification of the WT vs. open syntaxin signals normalized by β-tubulin revealed that there was no significant difference between the open mutant and WT levels (Supplementary Fig. S1d). Interestingly, we observed that the signal of open syntaxin was present not only at the 35 kDa monomer position but also at the ~80 kDa SNARE complex position despite that the worm sample was boiled once in SDS buffer (Supplementary Fig. S1c). The signal at ~80 kDa disappeared after repeated boiling of the samples. These results suggest that open syntaxin accelerates SNARE complex assembly as previously suggested[31].

To investigate the effects of the KI, we first assayed open syntaxin mutants for their motility and aldicarb sensitivity (Fig. 1). Aldicarb is an acetylcholinesterase inhibitor and has been widely used to examine the level of acetylcholine release at the C. elegans neuromuscular junction[35]. After 2 min in a liquid medium (M9 buffer), open syntaxin KI worms exhibited comparable ($p = 0.46$) thrashing rates to N2 WT worms (Fig. 1a). However, open syntaxin KI worms displayed a significantly faster onset of paralysis in the presence of aldicarb compared to N2 WT worms (Fig. 1b). To further examine whether open syntaxin

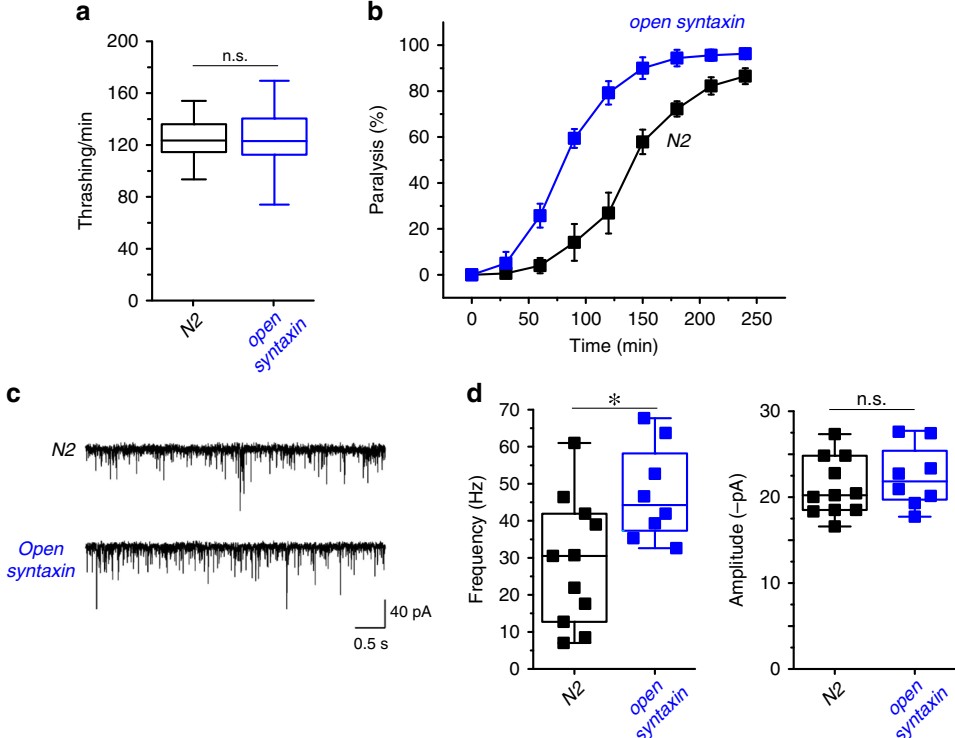

**Fig. 1 The knock-in open syntaxin mutation enhances exocytosis on a wild-type background. a** Box-and-whisker plots of motility for each strain, which was measured by counting the number of thrashes per min in M9 buffer. Knock-in open syntaxin *C. elegans* display similar thrashing rates as wild-type *C. elegans* (122 thrashes/min vs. 125 thrashes/min respectively). $n = 51$ for each strain. In two-sample two-sided *t*-tests, $t_{(100)} = 0.745$, $p = 0.46$. n.s. = not significant. **b** Aldicarb assays were conducted to measure aldicarb resistance of the animals. The *open syntaxin KI* mutant exhibits greater sensitivity to aldicarb compared to N2 wild-type worms. $n = 6$. Each assay was conducted with 15–20 worms. Error bars represent SEM. **c** Representative traces of mPSCs recorded from N2 (top) and *open syntaxin KI* (bottom) worms. **d** Box-and-whisker plots overlaid with the corresponding data points (squares) of mPSCs frequency (left) and amplitude (right) of N2 and *open syntaxin KI* worms. Two-sample two-sided *t*-test, *$p = 0.02$; n.s. $p = 0.44$. $n = 11$ for N2 and $n = 8$ for *open syntaxin* animals. Box-and-whisker plots represent the median (central line), 25th–75th percentile (bounds of the box), and 5th–95th percentile (whiskers).

increases synaptic vesicle release, we recorded miniature post-synaptic currents (mPSCs) at the *C. elegans* neuromuscular junction. We found a significantly increased frequency of mPSCs in the open syntaxin worms, while we did not observe statistically significant changes in the amplitude of mPSCs (Fig. 1c, d). Our results indicate that, compared to WT syntaxin, the KI open syntaxin mutation significantly enhances spontaneous exocytosis. The enhanced exocytosis of open syntaxin KI worms is likely explained by the accelerated SNARE complex assembly[31] (also Supplementary Fig. S1c). Thus, our results are different from those obtained by multicopy expression of open syntaxin in *unc-64*-null worms[25] and the phenotype of endogenously-expressed syntaxin KI animals is more consistent with the phenotype of syntaxin-1B KI mice[31,32].

**Open syntaxin partially rescues exocytosis defects in synaptotagmin-1 severe loss-of function mutants**. Together with previous results showing that multicopy expression of open syntaxin-1 partially rescued the exocytosis defects of *unc-13* and *unc-10* mutants[17,25], the increased spontaneous exocytosis caused by the KI open syntaxin mutation (Fig. 1) suggested the possibility that the open syntaxin mutation may provide a general means to enhance exocytosis and thus overcome defects caused by diverse types of mutations in the exocytotic machinery.

In this context, the ability of open syntaxin to rescue a null mutant of synaptotagmin-1/*snt-1*, a key calcium sensor for exocytosis[5,6], has not been investigated. We crossed a severe *snt-1* loss-of-function mutant allele, *snt-1(md290)*[36], with the open

syntaxin KI mutant to assay for thrashing ability and aldicarb sensitivity (Fig. 2). Importantly, we made the finding that open syntaxin can partially rescue synaptotagmin-1 mutant phenotypes in *C. elegans*. Thus, our *open syntaxin KI*; *snt-1(md290)* double mutants exhibited greater thrashing rates and paralysis in response to aldicarb than those observed for the *snt-1(md290)* mutant (Fig. 2a, b). Analysis of the mPSCs showed that open syntaxin increased the frequency of mPSCs but not the amplitude when compared to *snt-1* mutants alone (Fig. 2c, d). These results suggest that open syntaxin can bypass the requirement of a wider range of exocytosis regulators than previously recognized, including synaptotagmin.

**The open syntaxin KI mutant enhances evoked neurotransmitter release compared with WT syntaxin and partially restores the defects of evoked release of snt-1 mutants**. In *C. elegans* neuromuscular junctions, reliable recording of evoked neurotransmitter release by electrical stimulation has been a challenge. To overcome this technical difficulty, optogenetic stimulation has been developed in which Channelrhodopsin-2 is expressed specifically in cholinergic neurons (*zxIs6* strain) and evoked release is triggered by blue light[37].

To examine the effects of open syntaxin on evoked acetylcholine release, we generated *open syntaxin KI* and *open syntaxin KI*; *snt-1(md290)* mutants in the *zxIs6* background. When stimulated by 10 ms blue light (3.75 mW/mm$^2$), open *syntaxin KI* mutants exhibited a significant increase in the half-width and charge transfer of EPSCs compared to WT Fig. 3a, d, e). However, the

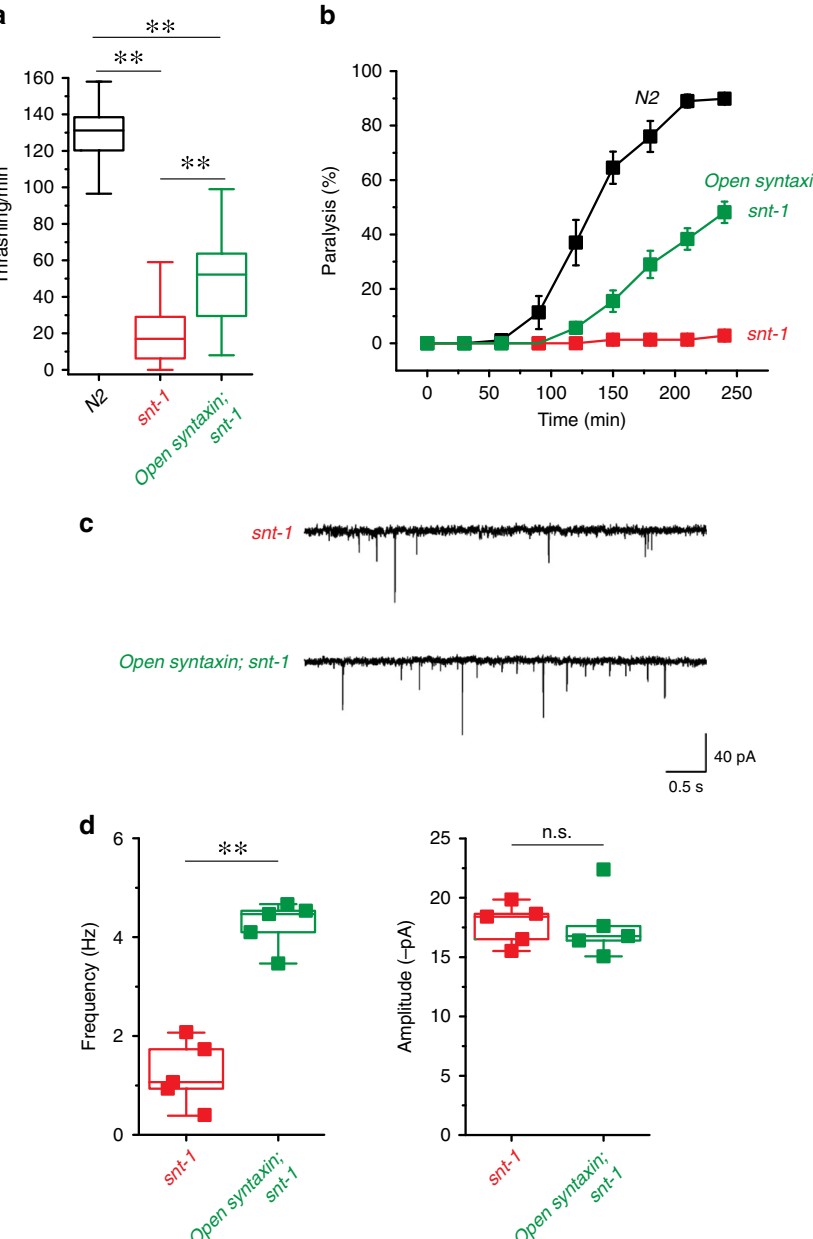

**Fig. 2 Open syntaxin rescues acetylcholine secretion defects in snt-1-null C. elegans. a** Box-and-whisker plots of thrashing assay for N2, *snt-1*, and *open syntaxin; snt-1* double mutants in M9 buffer. *snt-1* worms displayed impaired thrashing rates (20.1 thrashes/min), which was increased by the introduction of open syntaxin in the double mutant (48.9 thrashes/min). $n = 40$ for each strain. In one-way ANOVA statistical tests, $F_{(2,117)} = 388$ and $p = 0.00$. Tukey's test was performed for means analysis in ANOVA. N2 vs. *snt-1*: **$p = 0.00$, *snt-1* vs. *open syntaxin; snt-1*: $p = 0.00$, N2 vs. *open syntaxin, snt-1*: **$p = 0.00$. **b** Aldicarb assays of N2, *snt-1*, and *open syntaxin; snt-1* double mutants. *open syntaxin; snt-1* animals displayed slightly increased sensitivity to aldicarb compared to *snt-1* single mutants, which were resistant to aldicarb's paralyzing effects. $n = 6$. Each assay was conducted with 15–20 worms. Error bars represent SEM. **c** Representative mPSC traces recorded from *snt-1* and *open syntaxin; snt-1* worms. **d** Box-and-whisker plots overlaid with the corresponding data points (squares) of mPSC frequency (left) and amplitude (right) of *snt-1* and *open syntaxin; snt-1* worms. Two-sample two-sided t-test was performed. **$p = 0.00$; n.s. $p = 0.93$. $n = 5$ for *snt-1(md290)* and *open syntaxin; snt-1(md290)* animals. Box-and-whisker plots represent the median (central line), 25th–75th percentile (bounds of the box), and 5th–95th percentile (whiskers).

peak amplitude was unchanged (Fig. 3b). *Open syntaxin KI; snt-1 (md290)* mutants exhibited increases in the amplitude as well as the half-width and charge transfer of EPSCs compared with *snt-1 (md290)* animals (Fig. 3a, b, d, e). The rise time to the peak amplitude (10–90%) was unchanged in all the strains tested (Fig. 3c). Taken together, our results show that open syntaxin enhances not only the frequency of the spontaneous exocytosis (Figs. 1c, d and 2c, d) but also the amount of acetylcholine

released by light-induced depolarization. Furthermore, it partially rescues the defects of evoked exocytosis of *snt-1* mutant animals.

**Open syntaxin rescues exocytosis in unc-2 mutant C. elegans.** The key evidence that supported a specific genetic interaction between open syntaxin and UNC-13 derived from the observation that open syntaxin did not rescue exocytosis of the *unc-64*

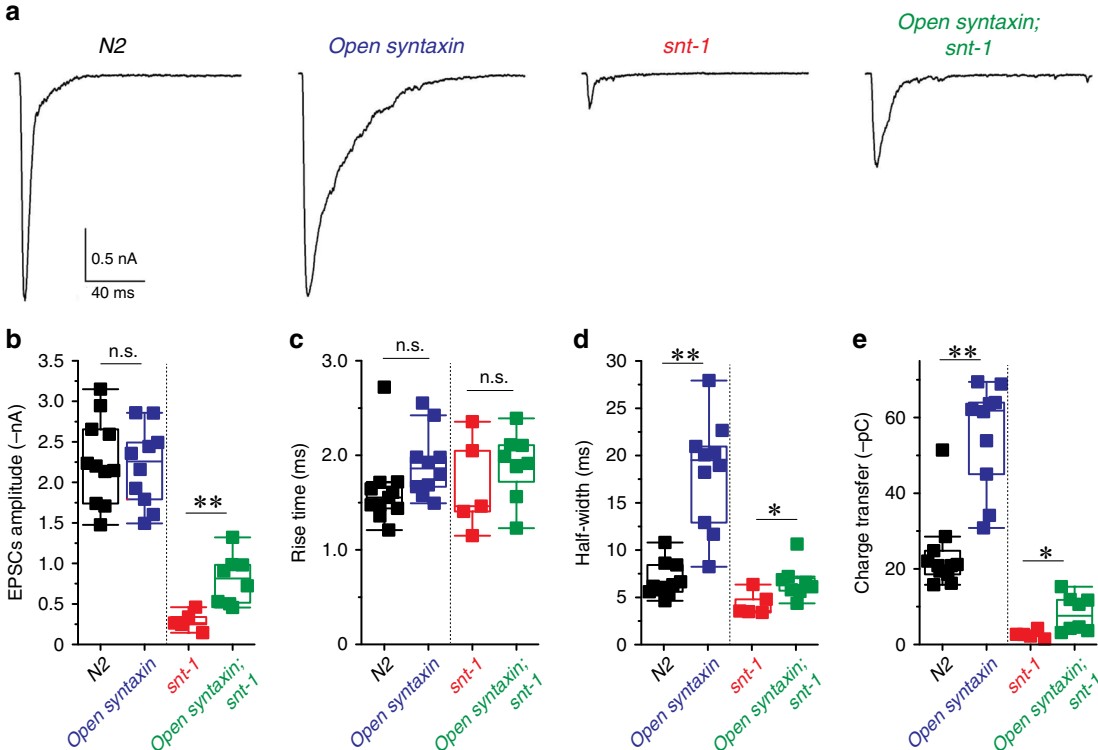

**Fig. 3 The knock-in open syntaxin mutation enhances evoked neurotransmitter release compared with the wild-type and partially restores the defects of evoked release of snt-1 mutants. a** Representative EPSCs traces recorded from wild-type N2, *open syntaxin*, *snt-1*, and *open syntaxin; snt-1* worms in the *zxIs6* transgenic background with 10 ms blue light illumination (3.75 mW/mm²). Muscle cells were recorded with a holding potential of −60 mV. **b–e** Box-and-whisker plots overlaid with the corresponding data points (squares) of EPSCs amplitude (**b**), rise time (10–90%) (**c**), half-width (**d**), and total charge transfer (**e**) of wild-type N2 (n = 11), *open syntaxin* (n = 10), *snt-1* (n = 5), and *open syntaxin; snt-1* (n = 8) worms, respectively. Two-sample two-sided *t*-test was performed for the comparison between WT N2 vs. *open syntaxin* and *snt-1* vs. *open syntaxin; snt-1* in each parameter. For amplitude (**b**), N2 vs. *open syntaxin*: n.s. $p = 0.75$, *snt-1* vs. *open syntaxin; snt-1*: **$p = 0.00$. For rise time (**c**), N2 vs. *open syntaxin*: n.s. $p = 0.10$; *snt-1* vs. *open syntaxin; snt-1*: n.s. $p = 0.39$. For half-width (**d**), N2 vs. *open syntaxin*: **$p = 0.00$, *snt-1* vs. *open syntaxin; snt-1*: *$p = 0.03$. For total charge transfer (**e**), N2 vs. *open syntaxin*: **$p = 0.00$; *snt-1* vs. *open syntaxin; snt-1*: *$p = 0.03$. Box-and-whisker plots represent the median (central line), 25th–75th percentile (bounds of the box), and 5th–95th percentile (whiskers).

(*js21*)-null mutant better than the WT syntaxin and did not rescue the reduced exocytosis caused by reduced Ca²⁺ influx in a severe loss-of-function mutant (*e55*) of UNC-2 Ca²⁺ channels[25]. UNC-2 is an α-subunit of P/Q/N-type Ca$_V$2-like voltage-gated calcium channels, which control the influx of calcium upstream of synaptic vesicle exocytosis[38]. The *e55* mutant has a Q571Stop mutation in the Ca$_V$2-like voltage-gated calcium channel α-subunit, which reduces calcium current and leads to impaired locomotion and exocytosis[38,39]. As our *open syntaxin KI*, or *unc-64 (sks4)*, increases synaptic exocytosis compared to WT N2 (Figs. 1 and 3) and partially rescues *snt-1* phenotypes (Figs. 2 and 3), we hypothesized that it might also rescue the reduced exocytosis of the *unc-2* loss-of-function mutant (*e55*).

*unc-2(e55)* worms exhibited reduced thrashing and aldicarb sensitivity, whereas *open syntaxin KI; unc-2(e55)* double mutants displayed robustly rescued thrashing ($p = 0.00$) compared to the *unc-2(e55)* mutant alone, and an aldicarb sensitivity comparable to the WT level (Fig. 4a, b). Analysis of the mPSCs showed that WT (N2), *unc-2(e55)*, and the *open syntaxin; unc-2(e55)* double mutants displayed similar amplitudes, but there was markedly reduced mPSC frequency in the *unc-2(e55)* worms that were rescued by *open syntaxin KI* in the double mutant (Fig. 4c, d). These striking results contrast with those obtained with multi-copy expression of open syntaxin-1[25] and provide compelling evidence that the open syntaxin mutation can at least partially overcome widely diverse defects in exocytosis.

We also tested the effect of open syntaxin on a gain-of-function allele of *unc-2(hp647)* (Supplementary Fig. S3). The *hp647* allele contains a L653F mutation[40]. A gain-of-function mutant allele of *unc-2*, *unc-2(hp647)* exhibited striking hypersensitivity to aldicarb (Supplementary Fig. S3b), suggesting that this mutant increased acetylcholine release. Interestingly however, thrashing frequency in *unc-2(hp647)* was not greater than that of the N2 WT (Supplementary Fig. S3a). *Open syntaxin KI; unc-2(hp647)* double mutants appeared to exhibit a slight further increase of aldicarb sensitivity when compared with *unc-2(hp647)* alone (Supplementary Fig. S3b), but it was difficult to manifest this effect using 1 mM aldicarb concentration. To further elucidate the difference between the strains, we used a lower aldicarb concentration plate (0.3 mM) and monitored for paralysis every 15 min (Supplementary Fig. S3c). We observed that *open syntaxin KI* slightly enhanced the aldicarb sensitivity of the *unc-2* gain-of-function mutant. Thus, open syntaxin can enhance exocytosis in both loss-of-function and gain-of-function mutants of the calcium channel UNC-2. These results support the possibility that open syntaxin increases synaptic transmission regardless of genetic background.

**Open syntaxin weakly enhances exocytosis in unc-13 mutants.** Unlike multicopy expression of open syntaxin in *unc-64*-null background, our KI open syntaxin enhanced exocytosis compared to the N2 WT (Fig. 1) and rescued exocytosis in *unc-2*

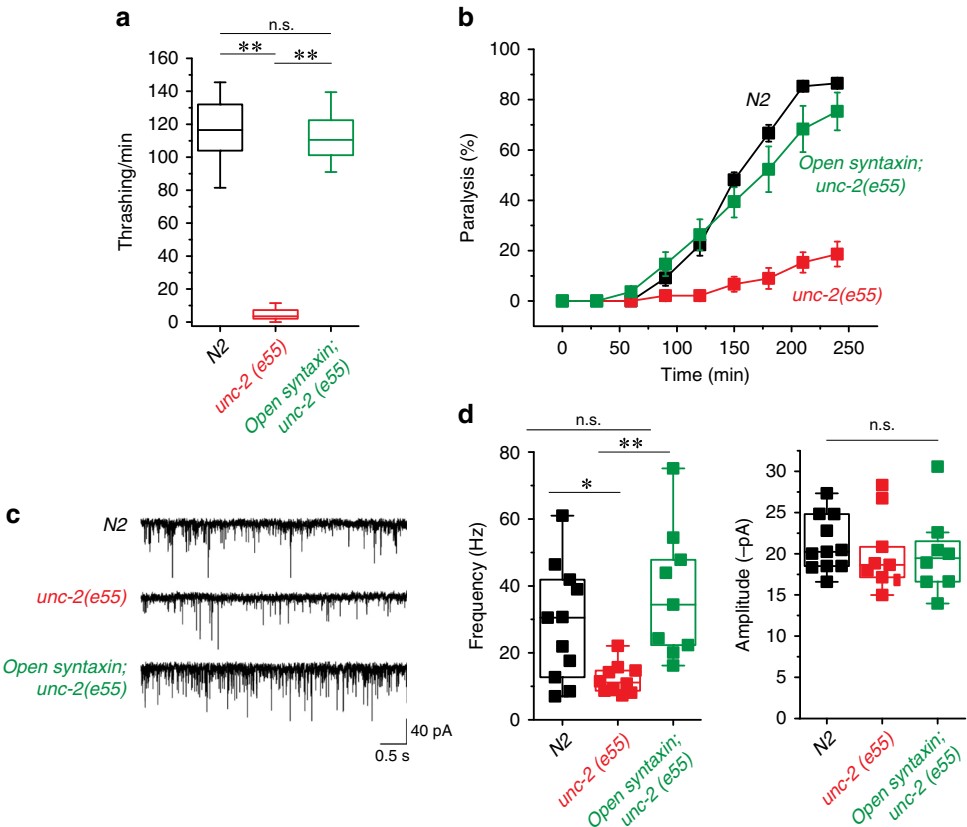

**Fig. 4 The knock-in open syntaxin mutation rescues the behavioral and exocytosis defects of *unc*-2-null *C. elegans*. a** Box-and-whisker plots of thrashing assay for N2, *unc-2(e55)*, and *open syntaxin; unc-2(e55)* double mutants in M9 buffer. *unc-2(e55)* worms displayed greatly reduced thrashing rates (5.43 thrashes/min), which was rescued by the introduction of open syntaxin in the double mutant. $n = 40$ for each strain. In one-way ANOVA statistical tests, $F_{(2,117)} = 836$ and $p = 0.00$. Tukey's test was performed for means analysis in ANOVA. N2 vs. *unc-2(e55)*: **$p = 0.00$; *unc-2(e55)* vs. *open syntaxin; unc-2 (e55)*: **$p = 0.00$, N2 vs. *unc-2(e55) open syntaxin*: n.s. $p = 0.13$. **b** Aldicarb assays of N2, *unc-2(e55)*, and *open syntaxin; unc-2(e55)* double mutants. *unc-2* animals displayed resistance to aldicarb, while the double mutant restored aldicarb sensitivity to near wild-type levels. $n = 6$. Each assay was conducted with 15–20 worms. Error bars represent SEM. **c** Representative mPSC traces recorded from N2, *unc-2(e55)*, and *open syntaxin; unc-2(e55)* worms. **d** Box-and-whisker plots overlaid with the corresponding data points (squares) of mPSC frequency (left) and amplitude (right) of N2, *unc-2(e55)*, and *open syntaxin; unc-2(e55)* worms. In one-way ANOVA tests, there was a significant difference among groups in frequency [$F_{(2,27)} = 7.01$ and $p = 0.00$]. Tukey's test was performed for means analysis in ANOVA for amplitude. For N2 vs. *unc-2(e55)*: *$p = 0.046$; for *unc-2(e55)* vs. *open syntaxin; unc-55(e55)*: **$p = 0.003$; for N2 vs. *open syntaxin; unc-2(e55)*: n.s. $p = 0.41$. There was no significant difference among groups in amplitude [$F_{(2,27)} = 0.23$ and $p = 0.79$]. $n = 11$ for N2, $n = 10$ for *unc-2(e55)* and $n = 9$ for *open syntaxin; unc-2(e55)* animals. Box-and-whisker plots represent the median (central line), 25th–75th percentile (bounds of the box), and 5th–95th percentile (whiskers).

loss-of-function mutant (Fig. 4). Therefore, we further investigated whether rescue of *unc-13* null *C. elegans* phenotypes by our open syntaxin KI mutation might differ from the results obtained with multicopy expression of open syntaxin[15,25]. We found that the severe thrashing defect of *unc-13(s69)* mutant (0.0125/min) was slightly alleviated by open syntaxin KI, which increased thrashing of *open syntaxin KI; unc-13(s69)* double mutant to 0.713/min (Fig. 5a). However, this effect was not statistically significant ($p = 0.87$) nor as strong as the effect of multicopy open syntaxin, which was reported to be ~6/min[25]. Moreover, we did not observe an increase in aldicarb sensitivity when compared to the *unc-13(s69)* mutant after the usual 4 h exposure (Fig. 5b). After 24 h in the aldicarb plate, ~33% of *open syntaxin KI; unc-13 (s69)* worms were paralyzed, compared to the 7% paralyzed *unc-13(s69)* worms. All WT worms paralyzed at the end of the 24 h. Thus, there is only a weak rescue of acetylcholine release from *unc-13(s69)* animals.

Previous studies using electrophysiological analyses reported conflicting levels of the rescue of synaptic vesicle exocytosis in *unc-13(s69)* animals by multicopy expression of open

syntaxin[15,25]. We therefore performed electrophysiological analyses of the effects of *open syntaxin KI* on *unc-13(s69)* mutants. The amplitudes of the mPSCs in the *unc-13(s69)* mutant were similar to those of the N2 worms, but the frequency of mPSCs were drastically reduced (Fig. 5i, j compare with Fig. 1c, d). The open syntaxin KI mutation on the *unc-13(s69)* background enhanced mPSC frequently significantly from <0.5 Hz to ~2 Hz (Fig. 5j). Nevertheless, the enhanced level did not reach WT levels (~30–40 Hz; Fig. 1d). To examine the effects of open syntaxin on evoked acetylcholine release of *unc-13(s69)* mutants, we generated *unc-13(s69)* as well as *open syntaxin KI; unc-13(s69)* mutants in the *zxIs6* background. When stimulated by 10-ms blue light, we found that although there was almost no evoked release in *unc-13 (s69)* mutant, open syntaxin significantly increased evoked release in the *unc-13(s69)* background (Fig. 5k). Yet the evoked release of the *unc-13; open syntaxin* double mutant is <10% of that of N2 WT and comparable with that of the open syntaxin; *snt-1* double mutant (Fig. 5l compare with Fig. 3). The weak rescue of *unc-13* phenotype by open syntaxin in electrophysiology is more consistent with the weak behavioral rescue observed in

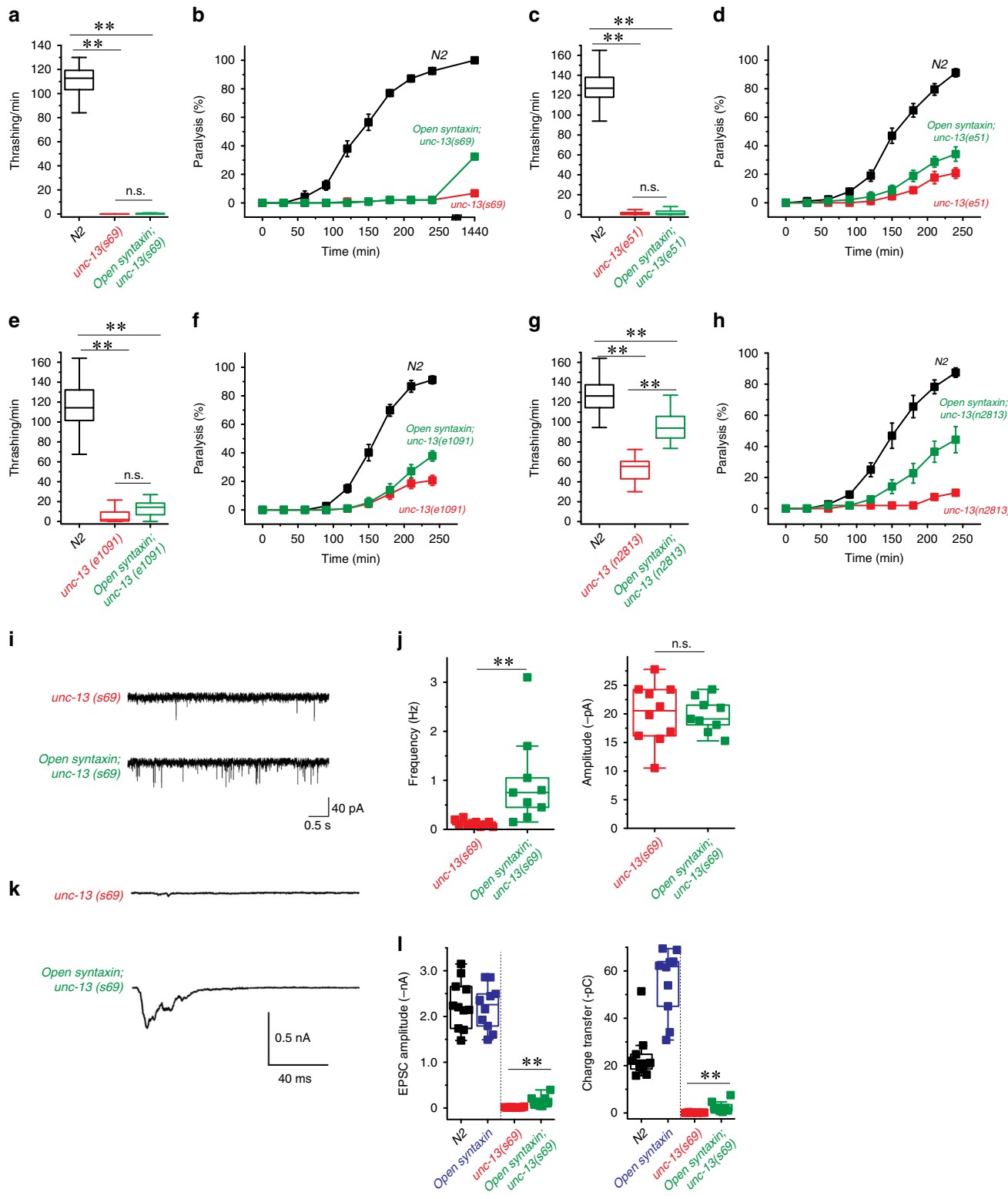

C. elegans[15] and the phenotype of Munc13-1/2 DKO neurons with overexpressed open syntaxin-1[33].

The UNC-13 protein exists as Left and Right region (LR) and Middle and Right region (MR) isoforms[12,41]. unc-13(s69) mutants possess a C1029fs mutation in both isoforms that results in a severe deleterious phenotype such that rescue with open syntaxin may be minimal. Therefore, we investigated whether the KI open syntaxin can rescue weaker loss-of-function alleles of unc-13: e51,

e1091, and n2813. e51 and e1091 mutants are null in their LR isoform only, while n2813 mutants possess a partial loss-of-function gene that results in a single S1574R missense mutation in both UNC-13 isoforms[12]. Open syntaxin KI; unc-13(e51) double mutants did not show rescued thrashing ($p = 0.91$) but did display increased sensitivity to aldicarb compared to the unc-13(e51) mutant alone (Fig. 5c, d). Similarly, open syntaxin KI; unc-13(e1091) double mutants did not show rescued thrashing

**Fig. 5 Open syntaxin weakly rescues the behavioral and exocytosis defects of *unc-13* mutants. a** Box-and-whisker plots of thrashing assay for N2, *unc-13 (s69)*, and *open syntaxin; unc-13(s69)* double mutants. $n = 40$ for each. One-way ANOVA, $F_{(2,117)} = 4270$ and $p = 0.00$. Tukey's test, N2 vs. *unc-13(s69)*: $**p = 0.00$, *unc-13(s69)* vs. *open syntaxin; unc-13(s69)*: n.s. $p = 0.87$, N2 vs. *open syntaxin; unc-13(s69)*: $**p = 0.00$. **b** Aldicarb assays of the indicated strains. $n = 6$ for 4 h assays, $n = 1$ for 24 h assay. Error bars represent SEM. **c** Thrashing of the indicated strains. $n = 57$ for each strain. One-way ANOVA, $F_{(2,168)} = 1760$ and $p = 0.00$. Tukey's test, N2 vs. *unc-13(e51)*: $**p = 0.00$; *unc-13(e51)* vs. *open syntaxin; unc-13(e51)*: n.s. $p = 0.91$, N2 vs. open syntaxin; *unc-13(e51)* $**p = 0.00$. **d** Aldicarb assays of the indicated strains. $n = 6$ for each. **e** Thrashing assay of indicated strains. $n = 40$ for each. One-way ANOVA, $F_{(2,117)} = 574$ and $p = 0.00$. Tukey's test, N2 vs. *unc-13(e1091)*: $**p = 0.00$, *unc-13(e1091)* vs. *open syntaxin; unc-13(e1091)*: n.s. $p = 0.13$, N2 vs. *open syntaxin; unc-13(e1091)* $**p = 0.00$. **f** Aldicarb assays of the indicated strains. $n = 6$. **g** Thrashing of the indicated strains. $n = 40$ for each. One-way ANOVA, $F_{(2,117)} = 173$ and $p = 0.00$. Tukey's test, For all comparisons, $**p = 0.00$. **h** Aldicarb assays of the indicated strains. $n = 7$. **i** Representative mPSC traces recorded from *unc-13(s69)* and *open syntaxin; unc-13(s69)* worms. **j** Summary data of mPSC frequency (left) and amplitude (right) of *unc-13(s69)* ($n = 10$) and *open syntaxin; unc-13(s69)* ($n = 9$) worms. Two-sample two-sided t-test, $**p = 0.0093$; n.s. $p = 0.92$. **k** Representative EPSCs traces recorded from *unc-13(s69)* and *open syntaxin; unc-13(s69)* worms in the *zxIs6* transgenic background. **l** Summary data of EPSCs amplitude and total charge transfer of *unc-13* ($n = 11$) and *open syntaxin; unc-13* ($n = 8$) worms were shown with the data of N2 and *open syntaxin* (adapted from Fig. 3 for comparison). Two-sample two-sided t-test for the comparison between *unc-13* and *open syntaxin; unc-13*. For amplitude, $**p = 0.00$, for charge transfer, $**p = 0.00$.

($p = 0.13$) but did exhibit increased sensitivity to aldicarb compared to the *unc-13(e1091)* alone (Fig. 5e, f). Finally, the *open syntaxin KI; unc-13(n2813)* double mutant displayed rescue in thrashing ($p = 0.00$) compared to the *unc-13(n2813)* mutant alone (Fig. 5g). Aldicarb sensitivity in the *open syntaxin KI; unc-13(n2813)* double mutant was also enhanced compared to the *unc-13(n2813)* mutant alone (Fig. 5h). Overall, our results suggest that open syntaxin increases acetylcholine release from *unc-13* mutants, but the level of rescue is rather limited and is less pronounced than the rescue of the *unc-2* loss-of-function mutant.

**The rescue of unc-13-null mutant with tom-1 null by open syntaxin is weaker than that by unfurled unc-18.** A limited rescue of *unc-13(s69)* by open syntaxin KI could be due to too severe impairment of synaptic vesicle priming in the absence of *unc-13*. In this context, the *tom-1* mutant that lacks Tomosyn, a protein that competes with synaptobrevin for SNARE complex formation[42], partially rescued *unc-13* phenotypes and multicopy expression of the open syntaxin mutant further increased the rescue to a modest extent[15]. In contrast, we recently discovered an interesting synergism between a KI of an *unc-18* mutant containing a point mutation (P334A) that is believed to unfurl a loop involved in synaptobrevin binding and the *tom-1(ok285)*-null mutant in rescuing the *unc-13(s69)* mutant in synaptic vesicle exocytosis[43]. That is, although *unc-18(P334A)* KI or *tom-1(ok285)* had a limited rescuing effect in aldicarb sensitivity of the *unc-13 (s69)* mutant, in combination they restored aldicarb sensitivity to the WT level. Therefore, we investigated whether introducing open syntaxin as a KI can restore *unc-13(s69)* strongly in the *tom-1* null background.

We first examined whether the effects of open syntaxin and *tom-1* on acetylcholine release and thrashing. As previously reported[15,43], *tom-1* null exhibited slightly reduced motility and thrashing compared to N2 WT, despite the increased acetylcholine release (Fig. 6a, b). Importantly, although open syntaxin KI did not affect the level of thrashing on *tom-1(ok285)* mutant, it further increased aldicarb sensitivity (Fig. 6b). These additive effects on aldicarb sensitivity suggest that facilitation of SNARE complex assembly by open syntaxin and the absence of TOM-1 can cooperate to further enhance exocytosis. In the triple mutants of *open syntaxin KI; tom-1 null; unc-13(s69)*, we observed that the open syntaxin partially increased motility and acetylcholine release of *unc-13* in the absence of *tom-1* (Fig. 6c, d). However, the rescue measured by the sensitivity to aldicarb is not as strong as that caused by the unfurled *unc-18(sks2)* mutant in the absence of TOM-1 and UNC-13. As we previously reported, the aldicarb sensitivity of unfurled *unc-18(sks2)* KI; *tom-1(ok285)*; *unc-13(s69)* reached N2 WT levels (Fig. 6f)[43]. These results suggest that *open syntaxin* and *tom-1* null exhibit a synergistic effect yet this

synergism is weaker than that of unfurled *unc-18* and *tom-1* null. We conclude that the ability of open syntaxin to suppress *unc-13* null is limited even in the absence of TOM-1.

**Rescue of unc-10 and unc-31 by open syntaxin KI.** Multicopy expression of open syntaxin-1, which can rescue the *unc-13(s69)* mutant, was also reported to rescue exocytosis defects of *unc-10* mutant[17] as well as docking defects of dense core vesicles of *unc-31* mutant. UNC-10 can bind and activate UNC-13[44–46]. UNC-31 is involved in dense core vesicle exocytosis but may also influence synaptic vesicle exocytosis[47,48]. To determine whether open syntaxin KI indeed enhances exocytosis of these synaptic exocytosis mutants, we crossed severe loss-of-function mutants, *unc-10 (md1117)* and *unc-31(e928) C. elegans*, with open syntaxin KI mutants to generate double mutants, and assayed for thrashing ability and aldicarb sensitivity (Supplementary Fig. S4). The *open syntaxin KI; unc-10(md1117)* double mutants displayed moderate rescue in thrashing rates ($p = 0.00$), as well as aldicarb sensitivity (Supplementary Fig. S4a, b). Similarly, *open syntaxin KI; unc-31 (e928)* double mutants also displayed moderate rescue of thrashing ($p = 0.00$), but restored aldicarb sensitivity to levels similar to those of WT worms (Supplementary Fig. S4c, d). These results further show that open syntaxin can suppress exocytosis defects from a wide range of synaptic transmission mutants. Furthermore, the degree of rescue of these mutants seems to be comparable to the rescue of *unc-13* mutants and in some cases is more effective.

**Open syntaxin aggravates the phenotype of unc-18 loss-of-function mutants.** Thus far, open syntaxin KI increased exocytosis to some degree in all the genetic backgrounds that we tested. Previous work suggested that multicopy over-expression of syntaxin does not bypass the requirement of UNC-18[9]; we thus sought to investigate whether open syntaxin KI can rescue *unc-18*-null worms. In the present study, we tested three strains of *unc-18* severe loss-of-function mutants: *md299*, *e81*, and *sks1*. The *md299* strain possesses a multigenic deletion resulting in the loss of the promoter and open reading frame of the *unc-18* gene[9]. The *e81* allele contains a C1582T mutation in exon 9, resulting in a Q530Stop mutation and is thus likely null of the *unc-18* gene[49]. Finally, the *sks1* strain contains a large insertion containing the genes for selection markers green fluorescent protein (GFP) and neomycin that disrupts proper transcription of the *unc-18* gene[43].

In all three *unc-18* alleles, thrashing rate was severely reduced and very little to no aldicarb sensitivity was observed (Fig. 7a, b and Supplementary Figs. S5a,b and S6a,b). Nevertheless, the *md299* strain exhibited measurable thrashing activity as previously reported[43]. Interestingly, *open syntaxin KI; unc-18*

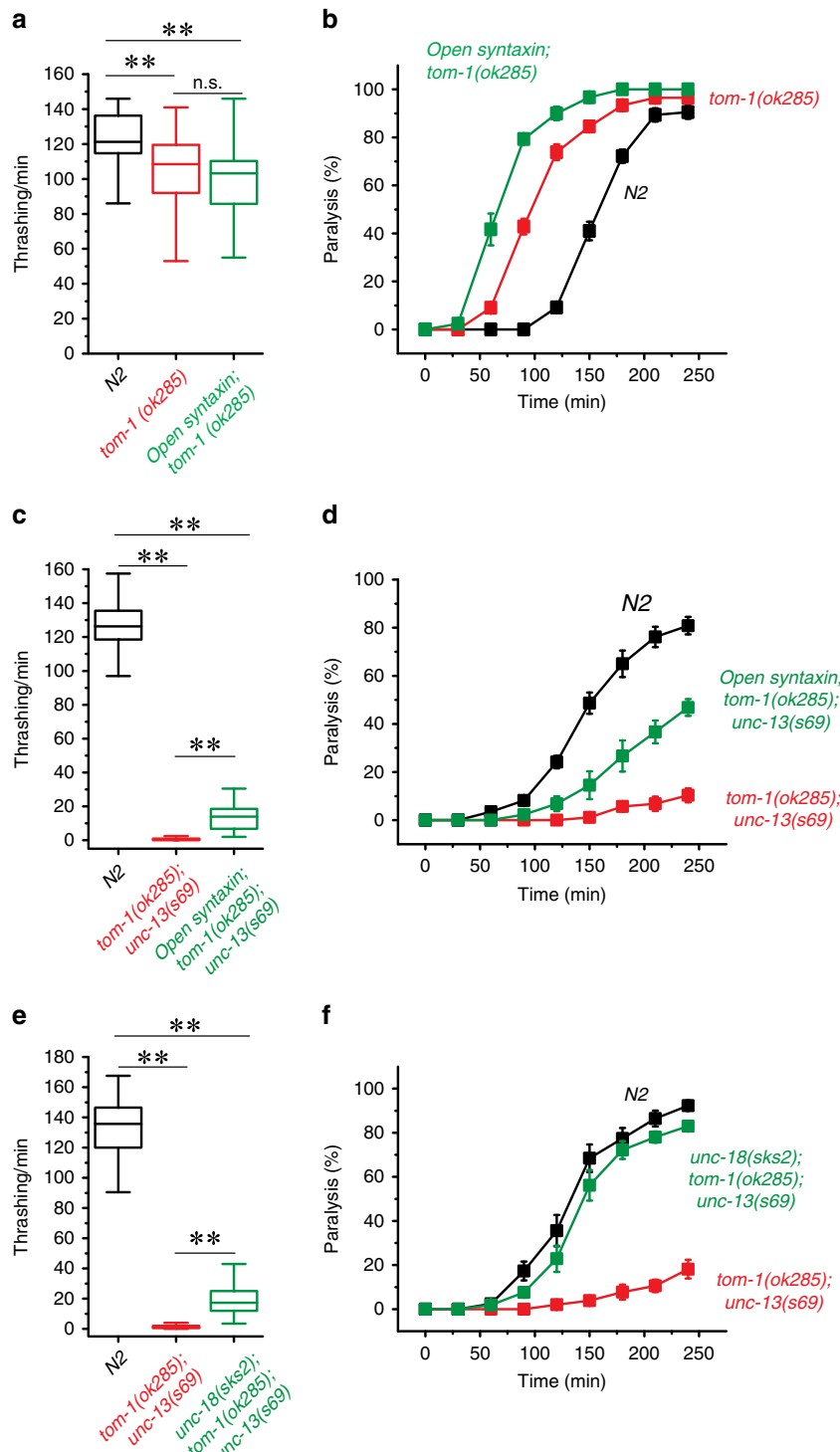

(md299) strongly reduced this thrashing activity (Fig. 7a). Thus, KI open syntaxin aggravated the motility defects of *unc-18* (md299). The double mutant and *md299* mutant displayed similarly poor aldicarb sensitivity. Therefore, worms were left overnight in the aldicarb plate and examined after 24 h for paralysis. At the end of the 24-hour period, 100% of WT worms were paralyzed, whereas ~7% of the *unc-18(md299)* worms were paralyzed, and double mutants did not paralyze on 1 mM aldicarb even after 24 h (Fig. 7b). The motility and aldicarb sensitivity of *e81* and *sks1* mutants were too low to detect a putative

aggravating effect of open syntaxin KI (Supplementary Figs. S5 and S6).

We examined the effects of open syntaxin KI on *unc-18* mutants using electrophysiology. *md299* mutants showed a reduced mPSC frequency compared to WT (compare Fig. 7c, d with Fig. 1c, d), and open syntaxin further reduced mPSC frequency in an *md299* background (Fig. 7c, d). The amplitude of mPSCs showed no difference between *md299* and *open syntaxin KI; md299* mutants. Despite our best efforts, *open syntaxin KI; unc-18(e81)* double mutants were too small and

**Fig. 6 Open syntaxin and unfurled *unc-18* differentially interact with *tom-1* in the rescue of *unc-13*. a** Box-and-whisker plots of thrashing assay for N2, *tom-1(ok285)* and *open syntaxin; tom-1* double mutants. $n = 40$ for each strain. In one-way ANOVA statistical tests, $F_{(2,117)} = 16.9$ and $p = 0.00$. Tukey's test was performed for means analysis in ANOVA. For N2 vs. *tom-1(ok285)*: \*\*$p = 0.00$; for *tom-1(ok285)* vs. *open syntaxin; tom-1(ok285)*: n.s. $p = 0.42$; for *N2* vs. *open syntaxin; tom-1(ok285)*: \*\*$p = 0.00$. **b** Aldicarb assays of N2, *tom-1(ok285)* and *open syntaxin; tom-1* double mutants. *open syntaxin; tom-1* animals displayed increased sensitivity to aldicarb compared to *tom-1* single mutants, while both displayed hypersensitivity to aldicarb compared to N2. $n = 6$. Each assay was conducted with 15–20 worms. Error bars represent SEM. **c** Thrashing assay of N2, *tom-1(ok285); unc-13(s69)*, and *open syntaxin; tom-1(ok285); unc-13(s69)* mutants in M9 buffer. The triple mutant rescued thrashing compared to the *tom-1(ok285); unc-13(s69)* double mutant thrashing rate. $n = 40$ for each strain. In one-way ANOVA statistical tests, $F_{(2,117)} = 2405$ and $p = 0.00$. Tukey's test was performed for means analysis in ANOVA. For all comparisons, \*\*$p = 0.00$. **d** Aldicarb assays of N2, *tom-1(ok285); unc-13(s69)*, and *open syntaxin; tom-1 (ok285); unc-13(s69)* mutants. *open syntaxin; tom-1 (ok285); unc-13(s69)* animals displayed increased sensitivity to aldicarb compared to *tom-1(ok285); unc-13(s69)* double mutants. $n = 6$. **e** Thrashing assay of N2, *tom-1(ok285); unc-13(s69)*, and *unc-18(sks2); tom-1(ok285); unc-13(s69)* mutants in M9 buffer. The triple mutant significantly rescued thrashing compared to the *tom-1(ok285); unc-13(s69)* double mutant thrashing rate. $n = 40$ for each strain. In one-way ANOVA statistical tests, $F_{(2,117)} = 1210$ and $p = 0.00$. Tukey's test was performed for means analysis in ANOVA. For all comparisons, \*\*$p = 0.00$. **f** Aldicarb assays of N2, *tom-1(ok285); unc-13(s69)*, and *unc-18(sks2); tom-1(ok285); unc-13(s69)* mutants. *unc-18(sks2); tom-1(ok285); unc-13(s69)* animals displayed increased sensitivity to aldicarb compared to *tom-1(ok285); unc-13(s69)* double mutants. $n = 7$. Box-and-whisker plots represent the median (central line), 25th–75th percentile (bounds of the box) and 5th–95th percentile (whiskers).

sick to perform reliable neuromuscular junction preparations for electrophysiology.

We indeed observed the striking effects of KI open syntaxin on the growth rate and body size of all the *unc-18* loss-of-function mutants. The slower growth rate and smaller size of *unc-18* single mutants were previously reported[49]. Starting from 15 eggs (day 0) in N2 WT worms, by day 7 the plate is populated with adults and the OP50 bacterial lawn is depleted. Because of their impaired movement, the various *unc-18* mutants were unable to fully populate the nematode growth media (NGM) plate and deplete the entire OP50 bacterial lawn. However, the areas of the plate where they were located were completely populated with smaller-sized adults and depleted of OP50 by day 9. In contrast, the *open syntaxin KI; unc-18* mutants failed to populate the plate or deplete the bacterial lawn to the same extent as the *unc-18* mutants by day 9 (Fig. 7e and Supplementary Figs. S5c and S6c). Our results strongly suggest that unlike all other synaptic transmission mutants, in which open syntaxin provides from modest to robust rescue, open syntaxin uniquely aggravates the phenotypes of *unc-18* null mutants. Thus, KI open syntaxin exhibits a specific genetic interaction with loss-of-function mutants of *unc-18*.

**The open syntaxin-1 partially bypasses the requirement of Munc-13-1 for liposome fusion.** Membrane fusion assays using reconstituted proteoliposomes provide a useful tool to correlate the physiological effects of mutations in components of the release machinery observed in vivo with the effects of these mutations in vitro on membrane fusion using minimal systems, yielding key insights into the mechanisms underlying the phenotypes[4,50]. Thus, the observation that fusion between liposomes containing synaptobrevin (V-liposomes) and liposomes containing syntaxin-1 and SNAP-25 (T-liposomes) in the presence of NSF and α-SNAP requires Munc18-1 and a Munc13-1 C-terminal fragment led to a model whereby Munc18-1 and Munc13-1 orchestrate SNARE complex assembly in an NSF-αSNAP-resistant manner, and provided a basis to understand the critical requirement of Munc18-1/UNC-18 and Munc13/UNC-13 for neurotransmitter release[51].

We used an analogous approach to examine how the open LE mutation that opens syntaxin-1 affects the requirement of Munc13-1 and Munc18-1 for liposome fusion, and thus test whether these reconstitution experiments correlate with the phenotypes of our genetic studies. Using an assay that simultaneously monitors lipid and content mixing[52], we observed fast, $Ca^{2+}$-dependent fusion between V-liposomes and T-liposomes containing WT syntaxin-1 in the presence of NSF,

α-SNAP, Munc18-1, a fragment spanning the $C_1$, $C_2B$, MUN, and $C_2C$ domains of Munc13-1 ($C_1C_2BMUNC_2C$), and a fragment spanning the $C_2$ domains that form the cytoplasmic region of synaptotagmin-1 ($C_2AB$) (Fig. 8a, b). Similar results were obtained in parallel experiments where the T-liposomes contained the open syntaxin-1 mutant, although we observed substantial $Ca^{2+}$-independent fusion. Importantly, no fusion was observed in the absence of Munc18-1 regardless of whether the T-liposomes contained WT or open mutant syntaxin-1 (Fig. 8a, b). Moreover, considerable $Ca^{2+}$-dependent fusion was observed in experiments performed with the syntaxin-1 open mutant where Munc18-1 was included but the Munc13-1 $C_1C_2BMUNC_2C$ fragment was absent, whereas practically no fusion was observed under these conditions with the liposomes containing WT syntaxin-1 (Fig. 8c, d and Supplementary Fig. S7). These results are reminiscent of those obtained with the Munc18-1 P335A mutant[43] and another Munc18-1 gain-of-function mutant (D326K)[53], and correlate with the partial rescue of *unc-13* phenotypes by the syntaxin-1 open mutation. Overall, these data nicely correlate with our in vivo experiments showing that the open syntaxin mutant can partially rescue *unc-13* phenotypes but not the phenotypes of *unc-18* mutants and are consistent with the notion that mutations that facilitate SNARE complex formation render membrane fusion and neurotransmitter release less critically dependent on Munc13/UNC-13.

**Open syntaxin animals do not significantly accelerate fusion pore kinetics of individual vesicles measured in the neuromuscular junction.** Despite the availability of many synaptic mutants, most mutants do not affect the fusion kinetics of synaptic vesicle fusion. Importantly, a recent study of open syntaxin-1B in mice showed that it significantly reduces rise time in miniature EPSCs (mEPSCs) in high-fidelity Calyx of Held synapses[32], whereas this effect was not observed in more conventional hippocampal synapses[31]. Therefore, we analyzed the kinetics of mPSCs of open syntaxin KI animals at the *C. elegans* neuromuscular junction. We found that there is no difference in rise time of mPSCs between WT control and open syntaxin KI animals (Supplementary Fig. S8a–c). In addition to two excitatory acetylcholine receptors, one inhibitory GABA receptor functions at the *C. elegans* neuromuscular junction[54]. The mPSC signals that we record are composed of both cholinergic mEPSCs and GABAergic miniature inhibitory postsynaptic currents (mIPSCs) and the mixture of two signals may interfere with the detection of subtle differences in fusion pore kinetics of individual vesicles.

To isolate cholinergic mEPSCs, we tested the effects of open syntaxin in the *unc-49(e407)* mutant, which lacks a functional

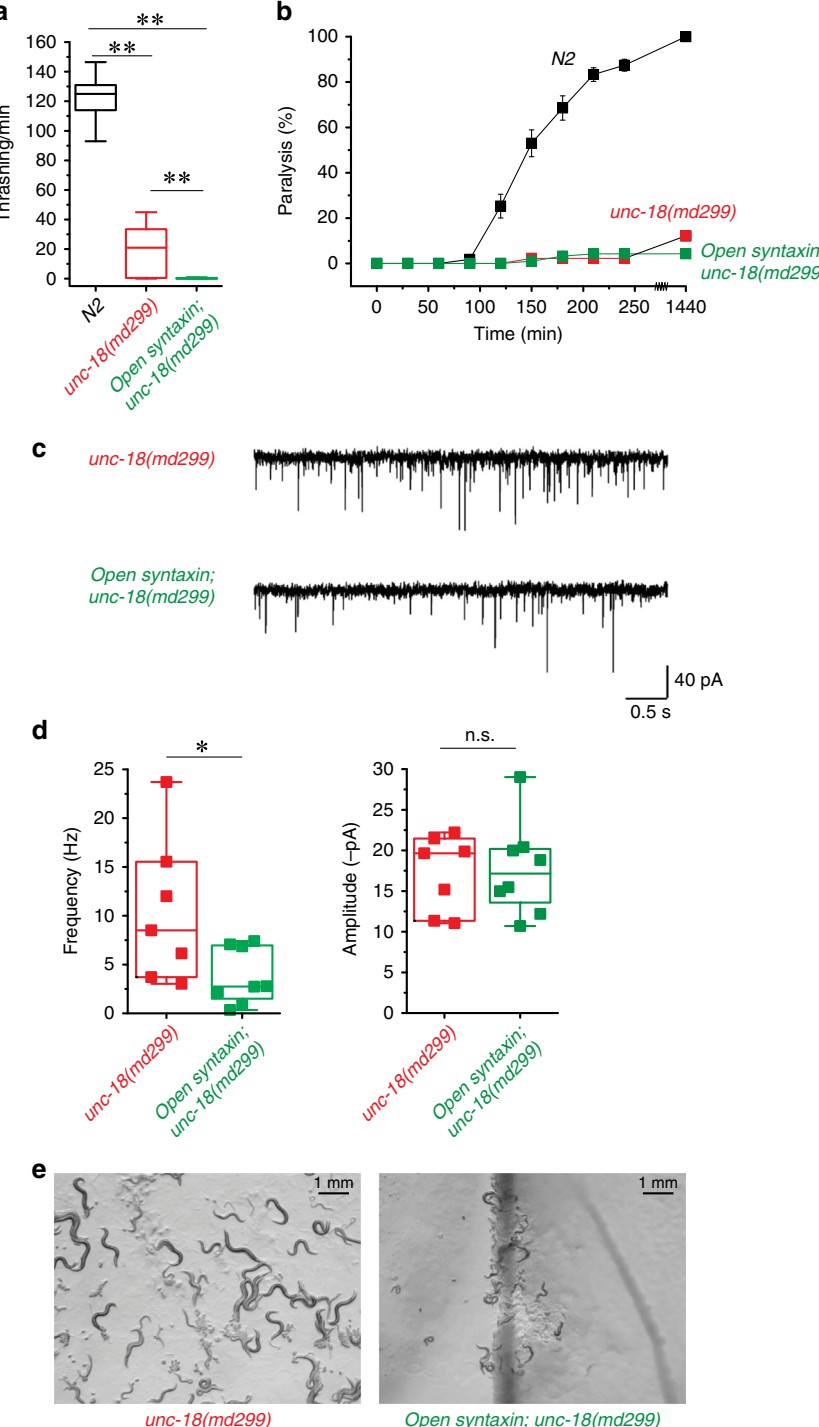

**Fig. 7 Open syntaxin further impairs the *unc-18* null phenotype. a** Box-and-whisker plots of thrashing assay for N2, *unc-18(md299)*, and *open syntaxin; unc-18(md299)* double mutants in M9 buffer. *unc-18(md299)* worms displayed greatly reduced thrashing rates (19.4 thrashes/min) which was reduced further by the introduction of open syntaxin in the double mutant to 0.610 thrashes/min. $n = 41$ for each strain. In one-way ANOVA statistical tests, $F_{(2,120)} = 1350$ and $p = 0.00$. Tukey's test was performed for means analysis in ANOVA. For all comparisons, **$p = 0.00$. **b** Aldicarb assays of N2, *unc-18(md299)*, and *open syntaxin; unc-18(md299)* double mutants. *unc-18* and *open syntaxin; unc-18* animals displayed similar resistance to aldicarb even after 24 h. $n = 7$ for 4 h assays, $n = 1$ for 24 h assay. Each assay was conducted with 15–20 worms. Error bars represent SEM. **c** Representative mPSC traces recorded from *unc-18 (md299)* and *open syntaxin; unc-18(md299)* worms. **d** Box-and-whisker graph overlaid with the corresponding data points (squares) of mPSC frequency (left) and amplitude (right) of *unc-18(md299)* and *open syntaxin; unc-18(md299)* worms. Two-sample two-sided *t*-test *$p = 0.04$; n.s. $p = 0.88$. $n = 7$ for *unc-18(md299)* and $n = 8$ for *open syntaxin; unc-18(md299)* animals. **e** Growth rate/body size images taken at day 9. Box-and-whisker plots represent the median (central line), 25th–75th percentile (bounds of the box), and 5th–95th percentile (whiskers).

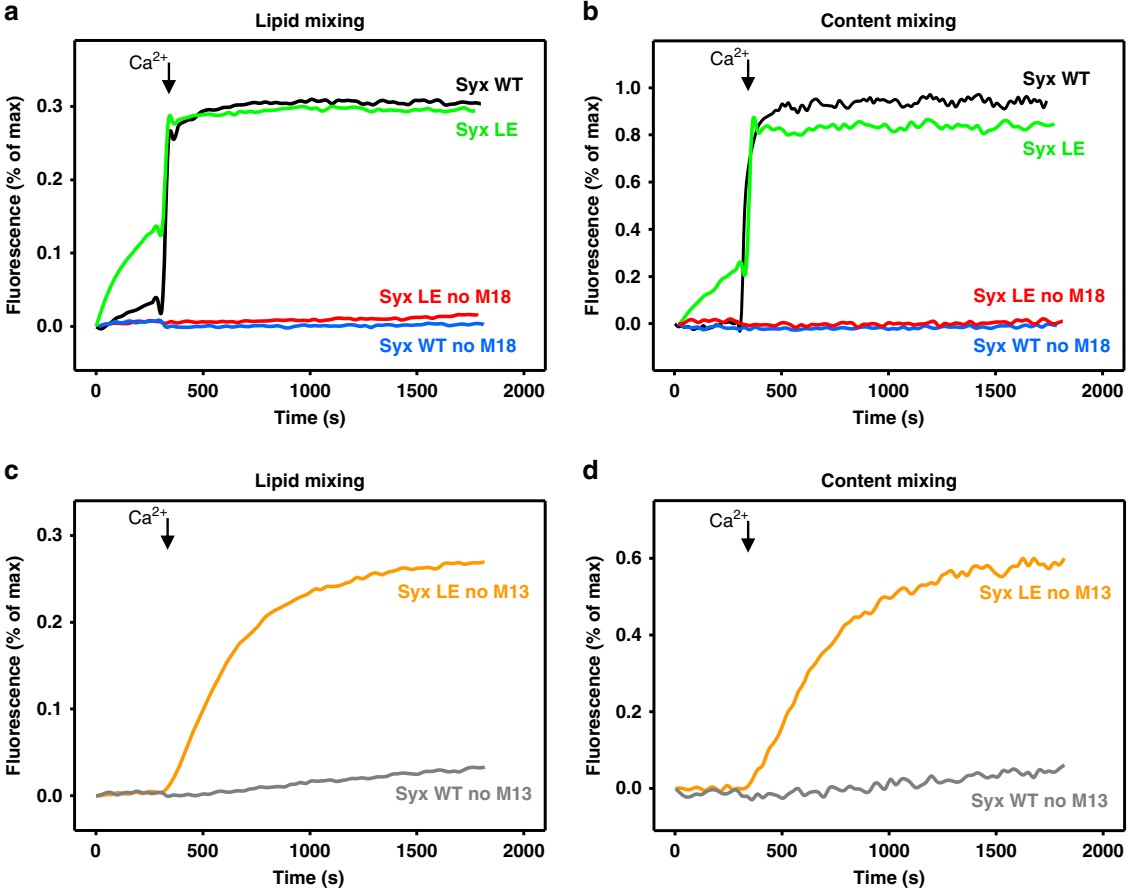

**Fig. 8 The syntaxin-1 open mutant partially bypasses the requirement of Munc13-1 but not Munc18-1 for liposome fusion. a–d** Lipid mixing (**a**, **c**) between V- and T-liposomes (containing syntaxin-1 (Syx) WT or LE open mutant) was monitored from the fluorescence de-quenching of Marina Blue lipids and content mixing (**b**, **d**) was monitored from the increase in the fluorescence signal of Cy5-streptavidin trapped in the V-liposomes caused by FRET with PhycoE-biotin trapped in the T-liposomes upon liposome fusion. In **a**, **b**, assays were performed in the presence of NSF, αSNAP, the synaptotagmin-1 $C_2AB$ fragment and the Munc13-1 $C_1C_2BMUNC_2C$ fragment with (green and black traces) or without (red and blue traces) Munc18-1. In **c**, **d**, assays were performed in the presence of NSF, α-SNAP, the synaptotagmin-1 $C_2AB$ fragment and Munc18-1, without the Munc13-1 $C_1C_2BMUNC_2C$ fragment. Experiments were started in the presence of 100 μM EGTA and 5 μM streptavidin, and $Ca^{2+}$ (600 μM) was added at 300 s.

$GABA_A$ receptor. In the thrashing assay, *unc-49*-null worms exhibited severely reduced motility (Fig. 9a). Interestingly, in the aldicarb assay, *unc-49* worms displayed hypersensitivity to aldicarb (Fig. 9b). We interpreted that the lack of $GABA_A$ receptor contributes to the lack of relaxation of muscle activity which enhances the aldicarb-induced paralysis while interfering with the thrashing behavior. We found that the open syntaxin further increased aldicarb sensitivity of *unc-49* worms (Fig. 9b), again suggesting that open syntaxin can increase transmitter release in many different genetic backgrounds. Electrophysiological recordings also indicated that open syntaxin induces a significant increase in frequency of cholinergic mEPSCs in the *unc-49(e407)* background (Fig. 9c, d). However, we did not find differences in the amplitude and rise times between *unc-49(e407)* mutants and *open syntaxin; unc-49(e407)* double mutants, as well as no differences in the kinetics of the mPSCs (Supplementary Fig. S8d–f). Therefore, our results suggest that open syntaxin KI does not enhance fusion pore kinetics of synaptic vesicles to a detectable level in conventional synapses.

## Discussion

Great advances have been made in the past three decades in our understanding of the mechanism of neurotransmitter release, showing that release depends on SNARE complexes that bridge the vesicle and plasma membranes, and that SNARE complex assembly is orchestrated by Munc18-1/UNC-18 and Munc13/UNC-13 (reviewed in ref. [4]). The tight complex between Munc18-1/UNC-18 and closed syntaxin serves as the starting point for this mechanism, and Munc13/UNC-13 was proposed to mediate the transition to the SNARE complex by opening syntaxin. A key finding that served as a basis for this model was that multicopy expression of the constitutively open LE syntaxin mutant[18] rescued the total abrogation of release observed in *C. elegans unc-13* nulls[25]. Subsequently, KI of open syntaxin-1B in mice was shown to enhance spontaneous release and vesicular release probability, which was proposed to arise from enhanced SNARE complex assembly, although open syntaxin-1B KI did not rescue the lethal Munc13-1 KO phenotype[31]. Our data now show that KI of open syntaxin in *C. elegans* enhances exocytosis in an otherwise WT background, as well as in a wide variety of genetic backgrounds characterized by diverse defects in synaptic transmission. These results indicate that the open syntaxin mutation provides a general means to enhance synaptic transmission and that increasing the number of SNARE complexes can overcome phenotypes that arise not only from defects in the SNARE complex assembly pathway but also from impairment of other aspects such of $Ca^{2+}$ sensing, $Ca^{2+}$ influx, or neurotransmitter recognition. These findings also suggest that manipulating SNARE complexes may provide a powerful tool for

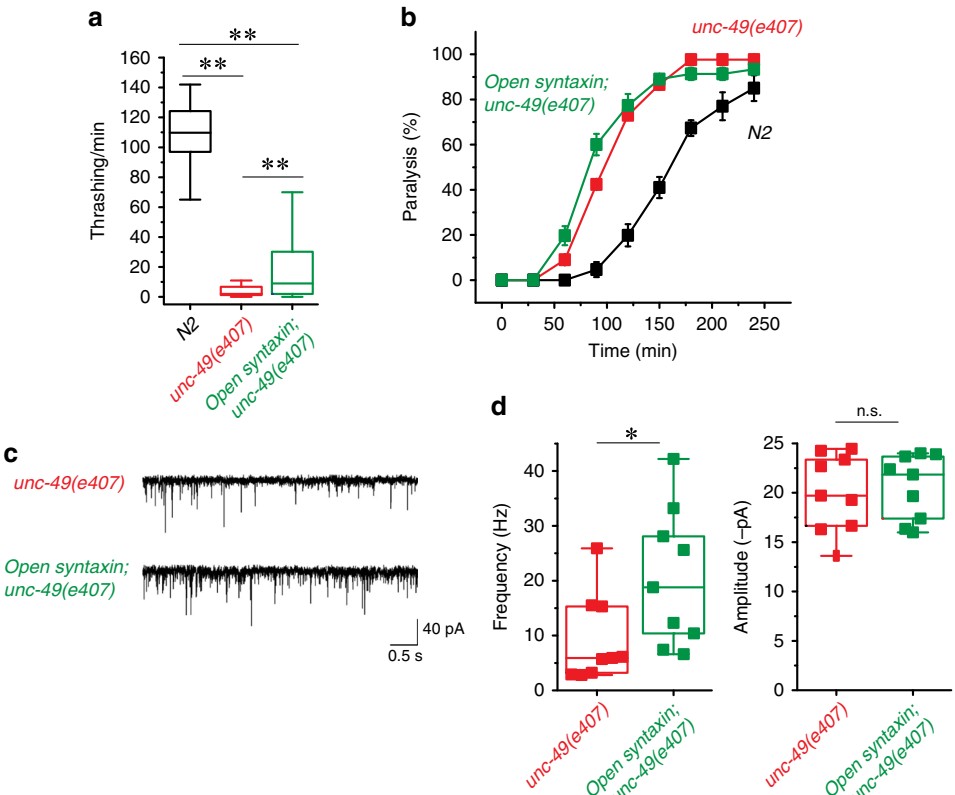

**Fig. 9 The knock-in open syntaxin mutation enhances exocytosis in GABA receptor mutants. a** Box-and-whisker plots of thrashing assay for N2, *unc-49 (e407)*, and *open syntaxin; unc-49(e407)* double mutants in M9 buffer. *unc-49(e407)* worms displayed greatly reduced thrashing rates (5.73 thrashes/min), which was rescued by the introduction of open syntaxin in the double mutant (17.05 thrashes/min). $n = 40$ for each strain. In one-way ANOVA statistical tests, $F_{(2,117)} = 335$ and $p = 0.00$. Tukey's test was performed for means analysis in ANOVA. For all comparisons, $**p = 0.00$. **b** Aldicarb assays of N2, *unc-49(e407)*, and *open syntaxin; unc-49(e407)* double mutants. *unc-49* animals displayed hypersensitivity to aldicarb (gray), which was further enhanced in the double mutant (black). $n = 6$. Each assay was conducted with 15–20 worms. Error bars represent SEM. **c** Representative mEPSC traces recorded from N2, *unc-49(e407)*, and *open syntaxin; unc-49(e407)* worms. **d** Box-and-whisker plots overlaid with the corresponding data points (squares) of mEPSC frequency (left) and amplitude (right) of N2, *unc-49(e407)*, and *open syntaxin; unc-49(e407)* worms. Two-sample two-sided *t*-test, $*p = 0.04$; n.s. $p = 0.75$. $n = 9$ for *unc-49(e407)* and *open syntaxin; unc-49(e407)* animals. Box-and-whisker plots represent the median (central line), 25th–75th percentile (bounds of the box), and 5th–95th percentile (whiskers).

functional studies of synaptic transmission and neural networks, as well as opportunities to develop therapies for diverse neurological disorders.

The pioneering studies of Richmond et al.[25] provided crucial evidence demonstrating the critical functional importance of opening syntaxin in a physiological context. Although some of the conclusions might now be questioned, this work paved the way for a flourish of additional studies that investigated genetic interactions involving opening of syntaxin. The observation that multicopy expression of open syntaxin rescued not only *unc-13* but also of *unc-10* and *unc-31* phenotypes[17,30] suggested that more than one protein might be involved in opening syntaxin and/or that Munc13/UNC-13 might have only an indirect role in this event. However, RIM/UNC-10 is known to play a key role in synaptic vesicle priming by activating Munc13-1/UNC-13[46], and the functions of CAPS/UNC-31, which contains a MUN domain homologous to that of Munc13/UNC-13, are likely related to those of Munc13/UNC-13 (reviewed in ref. [55]). Hence, these results were not incompatible with the proposed role of Munc13/UNC-13 in opening syntaxin.

Using an open syntaxin KI approach that does not suffer from technical problems that can arise from multicopy expression, our data now show that (i) *open syntaxin KI* increases synaptic vesicle exocytosis in an otherwise WT background (Fig. 1); (ii) *open syntaxin KI* can increase neurotransmitter release in a wide range

of synaptic mutants that include *snt-1* (Figs. 2 and 3), *unc-2* (Fig. 4 and Supplementary Fig. S3), *tom-1* (Fig. 6), *unc-31* (Supplementary Fig. S4), and *unc-49* (Fig. 9), in addition to the previously reported *unc-13* (Fig. 5) and *unc-10* (Fig. S4); and (iii) the rescue of various *unc-13* mutants (Fig. 5) by open syntaxin KI is rather weak and contrasts with the stronger rescues observed for some of the other mutants. The gain-of-function caused by open syntaxin KI in *C. elegans* is reminiscent of the increased spontaneous release and vesicular release probability observed in open syntaxin-1B KI mice[31]. Conversely, we did observe an increase in the charge transfer of evoked release in the open syntaxin KI worms (Fig. 3e), which was not observed in open syntaxin-1B KI mice[31]. Further research will be needed to assess whether this difference arises from distinct impairment of the RRP, which was decreased in the KI mice[31]. Regardless of this difference, the gain-of-function phenotypes caused by the open syntaxin mutation in mice and worms are most likely associated with the fact that this mutation facilitates SNARE complex assembly[18] and an increase in the number of assembled SNARE complexes enhances the probability of vesicle release[32]. Although the importance of SNARE complex assembly for neurotransmitter release has been recognized for a long time, it is remarkable that open syntaxin can rescue the defects caused by ablation of proteins with functions as diverse as those of the $Ca^{2+}$ sensor synaptotagmin, a $Ca^{2+}$ channel and a $GABA_A$ receptor.

Why can open syntaxin almost fully rescue the *unc-2(e55)* null mutant (Fig. 2), while its rescue of synaptic mutants such as *snt-1 (md290)* (Figs. 1 and 3) or *unc-13(s69)* (Fig. 5) is limited? We suggest that there is still some calcium influx in the *unc-2(e55)* null mutant via the other $Ca^{2+}$ channels, namely EGL-19, and we propose that the *open syntaxin-1* mutant overcomes the reduced $Ca^{2+}$ influx in *unc-2(e55)* mutant by increasing the $Ca^{2+}$ sensitivity of the release machinery, which was actually shown in Gerber et al.[31]. There are at least two additional $Ca^{2+}$ channels (EGL-19, L-type; CCA-1, T-type) in addition to UNC-2 (P/Q/N-type) in the genome of *C. elegans*[56]. Importantly, Liu et al.[57] showed that SLO-2 $K^+$ channels regulate neurotransmitter release from cholinergic and GABAergic motor neurons and that the activity of SLO-2 in motor neurons depends on $Ca^{2+}$ entry through EGL-19, but not other $Ca^{2+}$ channels (CCA-1 or UNC-2). Thus, EGL-19 $Ca^{2+}$ channels play a regulatory role in neurotransmitter release from motor neurons, although direct evidence of EGL-19 in neurotransmitter release had been difficult to obtain due to the lethality of *egl-19* null mutant[58]. Therefore, although UNC-2 is the major $Ca^{2+}$ channel that controls neurotransmitter release, EGL-19 seems to play an additional role. This also explains why the null mutant of *unc-2(e55)* still shows significant residual acetylcholine release and motility (see Fig. 2). We further suggest that open syntaxin overcomes the lack of UNC-2 by increasing the sensitivity of the release machinery to residual $Ca^{2+}$ influx from EGL-19 channels. Partial rescue of snt-1-null mutant may be explained by the presence of several synaptotagmin isoforms (SNT-2, SNT-3, etc. in wormbase) in the genome of *C. elegans*. These isoforms may replace SNT-1 and function similarly but not very efficiently, and an increased number of SNARE complexes by open syntaxin can compensate for this defect.

The limited rescue of *unc-13* phenotypes by the open syntaxin KI (Fig. 5) correlates with studies that observed no rescue or much weaker rescues in *C. elegans unc-13*[15,59] and in Munc13-deficient mice[31,33] than those reported by Richmond et al.[25]. One might argue that such limited rescues arise because of the very strong nature of the *unc-13* phenotypes, which constitutes a strong challenge for any type of rescue. However, the rescue afforded by open syntaxin KI is rather modest even when combined with the *tom-1(ok285)* mutation (Fig. 5), which alleviates the severity of the *unc-13* phenotypes. Moreover, open syntaxin KI produces considerably stronger rescues of other mutants (e.g., *unc-2*, Fig. 4) and the P334A mutation in UNC-18 rescues the aldicarb sensitivity of the *unc-13(s69)* mutant better than open syntaxin (Fig. 6). Based on these findings, one might question whether Munc13/UNC-13 actually has a direct role in opening syntaxin. However, biochemical and biophysical data have shown that the Munc13-1 MUN domain accelerates the transition from the Munc18-1/closed syntaxin complex to the SNARE complex[27], most likely through direct interactions with syntaxin[28,29]. A plausible explanation for the apparent contradiction between these findings and the limited rescue of *unc-13* phenotypes by open syntaxin is that opening syntaxin is one of the roles of Munc13/UNC-13 but is not the primary function or at least not the sole primary function. In this context, the large, conserved C-terminal region of Munc13-1 has been shown to have a highly elongated structure[60] that has been proposed to provide a bridge between the plasma membrane and synaptic vesicles[52]. Strong evidence supporting this model has been recently provided by the observation that even a single point mutation in this 200 kDa protein that abolishes its membrane–membrane bridging ability abrogates neurotransmitter release almost completely[61]. It is also worth noting that Munc13-1 has been shown to improve the fidelity of SNARE complex assembly, preventing antiparallel orientations of the SNAREs[33].

Interestingly, multicopy expression of open syntaxin cannot rescue *unc-18* phenotypes[9], and in our experiments with open syntaxin KI the phenotypes of all the mutant strains that we tested were improved, except for *unc-18* mutants. In fact, open syntaxin KI aggravates *unc-18* phenotypes (Fig. 7 and Supplementary Figs. S5 and S6). This lack of rescue in the absence of UNC-18 and the partial rescue in the absence of UNC-13 are nicely mirrored in our reconstitution assays, where the open syntaxin-1 mutant enables some liposome fusion in the absence of Munc13-1 $C_1C_2BMUNC_2C$ (unlike WT syntaxin-1), but not in the absence of Munc18-1 (Fig. 8). These findings can be readily explained by results from reconstitution assays showing that Munc18-1 and Munc13-1 are normally essential to form *trans*-SNARE complexes in the presence of NSF and αSNAP, because Munc18-1 and Munc13-1 orchestrate SNARE complex assembly in an NSF-αSNAP-resistant manner[51,52,62]. The requirement of Munc13-1 for liposome fusion can be partially bypassed by the open syntaxin mutation (Fig. 8) or by mutations in Munc18-1 that also facilitate SNARE complex assembly[43,53]. However, Munc18-1 cannot be bypassed, likely for three reasons. First, because Munc18-1 provides a template to assemble the SNARE complex[53,63,64]. Second, because αSNAP completely blocks SNARE-dependent liposome fusion through diverse interactions with the SNAREs and such inhibition can only be overcome by binding of Munc18-1 to closed syntaxin-1, suggesting that this complex constitutes an essential starting point for the productive pathway that leads to neurotransmitter release[65]. Thus, the observation that the open syntaxin mutation exacerbates rather than alleviates the *Unc18* phenotypes can be rationalized by the impairment of the interaction of syntaxin with Unc18/Munc18-1 caused by this mutation. A third reason is that anterograde trafficking of syntaxin/UNC-64 is known to be severely impaired in *unc-18* mutants[21] and similar defects have been demonstrated in Munc18-1/2 double knockdown PC12 cells[19,20,66,67], as well as in Munc18-1 KO neurons[22]. This trafficking defect is believed to be due to the formation of ectopic SNARE complexes in the ER and Golgi[68]. Such ectopic SNARE formation is accelerated when syntaxin/UNC-64 adopts the open conformation, which may contribute to the observed aggravation of *unc-18* phenotypes.

Does open syntaxin enhance fusion pore opening kinetics of individual vesicles? Several mechanisms to control fusion pore opening have been suggested, including the copy number of SNARE complexes. Nevertheless, there were almost no synaptic mutants known that affect the fusion kinetics of synaptic vesicle fusion. Using Calyx of Held synapses in mice, Acuna et al.[32] demonstrated that open syntaxin-1B can reduce rise time of mEPSCs. The results appear to be consistent with a recent study showing that cleavage of synaptobrevin by tetanus toxin reduces the amplitude and kinetics of fusion events in cultured neurons[69]. We did not observe changes in amplitude or kinetics of mEPSCs in open syntaxin KI despite a significant increase in frequency of mEPSCs. One potential explanation is that, in the Calyx of Held, the rise time is very fast (~0.1 ms), which may imply that the diffusion time of neurotransmitter glutamates to reach receptors is short, allowing detection of small changes in the kinetics of vesicle fusion. The rise time in *C. elegans* neuromuscular junctions is significantly slower at ~0.6 ms. However, the effects of synaptobrevin cleavage by tetanus toxin on the amplitude and kinetics were clearly observed in hippocampal cultured neurons in which the rise time is 1-2 ms[69]. It might be possible to find synaptic mutants which can slow down the kinetics of vesicle fusion events at the appreciable level in *C. elegans* in the future.

Taken together, our results show that open syntaxin can suppress a wide range of exocytosis defects but not those of Munc18/UNC-18 mutants. Recent studies show that mutations in syntaxin-1B, Munc18, Munc13, and Tomosyn underlie a wide spectrum of childhood epilepsy and autism spectrum disorders (ASDs)[70–78]. Similarly, mutations in proteins involved in the

trafficking of postsynaptic neurotransmitter receptors such as Shank3[79–81], and neuroligin-3 and -4[82] are strongly implicated with ASD. Thus, small chemicals that facilitate opening of syntaxin-1 have the potential to alleviate many neurological diseases derived from defects in synaptic transmission by increasing synaptic transmitter release.

## Methods

**CRISPR-mediated genome editing of unc-64.** The strategy is shown in Supplementary Fig. S1. First, a two-step PCR was performed to generate two different PCR products encoding two different guide RNA (sgRNAs) that would target 5′-gCTGTACCTGCCTACAAGGcgg-3′ sequence and 5′-AAGACGAACCCAGAGAACATcgg-3′ sequence within the intron of unc-64 genomic region (see Supplementary Fig. S2). Second, a repair vector was constructed that contained a dual-marker selection cassette (GFP and neomycin-resistant gene[34] flanked by left and right homology arms of unc-64. Upstream (left arm) and downstream (right arm) sequences of the CRISPR target site were amplified by PCR using pTX21 (a kind gift from Dr. Michael Nonet), which contains the genomic sequence of the unc-64 gene[26]. The left arm PCR product (~1.1 kb) was digested with SacI and NotI, whereas the right arm PCR product (~1.7 kb) was digested with SpeI and ClaI. Each of the digested products was ligated into pBluescript, followed by sequencing of the ligated products. To generate a KI mutant allele, site-directed mutagenesis was conducted on pBluescript containing the right arm. The mutagenesis product was verified by sequencing. The dual-marker selection cassette (~5.4 kb) was digested with SpeI and NotI and ligated into pBluescript containing both arms[34]. Third, multiple plasmids were injected into N2 animals: a vector encoding Cas9, the repair vector, the DNA product of sgRNA, and several injection markers (Pmyo-3-mCherry and Prab-3-mCherry). Fourth, screening for KI animals began by applying neomycin to the progeny of the injected animals. Then, animals that were neomycin resistant, GFP positive, and mCherry negative were selected as KI candidates and genotyped for verification, resulting in the selection of unc-64(sks3) worm. We then injected Cre-recombinase gene into unc-64(sks3) worms to remove GFP and neomycin-resistant gene. The genomic sequence of the resulting worm unc-64(sks4) was shown in Supplementary Fig. S2.

**Genetics.** All strains used in the study were maintained at 22 °C on 30 mm agar NGM plates seeded with OP50, a strain of Escherichia coli, as a food source. The C. elegans strains used were listed in Table 1. unc-10(md1117), unc-13(e51), (e1091), (n2813), unc-31(e928), unc-2(e55), unc-18(md299), and snt-1(md290) worms were purchased from the Caenorhabditis Genetics Center (University of Minnesota); VC223 strain with a genotype of tom-1(ok285) was provided by the C. elegans

Reverse Genetics Core Facility at the University of British Columbia; and unc-2 (hp647) was generated in Mei Zhen's lab[40]. We crossed these mutants with unc-64 (sks4) to generate double and triple mutants. Strains with zxIs6 background were cultured in the dark at 22 °C on OP50-seeded NGM plates supplemented with all-trans retinal (0.5 mM). Fifteen to 20 adult worms from each plate were dissolved in worm lysis buffer to extract their DNA for PCR. PCR was conducted to confirm the genotype of double or triple mutants using primers purchased from IDT DNA. PCR primers used to verify unc-64(sks4) LE open mutant allele in comparison with WT allele are SS1727 (5′-GGTGTAAGGGACGAATTCAGAG-3′) and SS1728 5′-CAAACCTGTTGGCTATCTGTGA-3′).

**Western blot analysis of UNC-64 expression in C. elegans worms.** Protein extract was prepared by culturing N2 and unc-64(sks4) worms on 3.5 or 6 cm NGM plates. When the plates were full of worms with little OP50 left, worms were harvested and washed three times with a buffer containing 360 mM sucrose and 12 mM HEPES. These worms were resuspended in ~5 times volume of the buffer and frozen at −80 °C until use. The defrosted worms were sonicated on ice 10 times with a 5 s burst. The resulting lysate was centrifuged for 15 min to pellet the cuticle, nuclei, and other debris. After centrifugation, the supernatant (final protein concentrations were ~3–5 mg/ml) was transferred to a clean microcentrifuge tube with an equal volume of sample buffer (2×). Fifty micrograms of samples were subjected to SDS-polyacrylamide gel electrophoresis followed by immunoblotting. The immunoblot signal was detected with ECL Prime Western Blotting Detection Reagent (GE Healthcare) and a Chemidoc XRS + Imaging System (Bio-Rad), and quantified by ImageJ program. After the first round of immunoblotting to detect UNC-64 with rabbit polyclonal anti-syntaxin-1 antibody (I378, 1:1000)[18], the membrane was re-used for a second immunoblot assay with a mouse monoclonal anti-β-tubulin antibody (E7 clone from Developmental Studies Hybridoma Bank, 1:2000) to correct for sample loading.

**Behavioral analyses.** Motility of each strain was determined by counting the thrashing rate of C. elegans in liquid medium. Briefly, worms were bleached to release their eggs and were then synchronously grown to young adulthood. Young adult worms were placed in a 60 μL drop of M9 buffer on a 30 mm petri dish cover. After a 2 min recovery period, worms were video-recorded for 2 min using an OMAX A3580U camera on a dissecting microscope with the OMAX ToupView v3.7 software. The number of thrashes per minute were manually counted and averaged within each strain. A thrash was defined as a complete bend in the opposite direction at the midpoint of the body. At least 40 worms for each strain was measured in each analysis.

## Table 1 C. elegans knock-in mutants, alleles, and strains.

| Genotype | Strain # |
|---|---|
| unc-64(sks3[L166A/E167A + loxP Pmyo-2-GFP; Prps-27-NeoR loxP) III | UHN28 |
| unc-64(sks4[L166A/E167A + loxP]) III | UHN29 |
| unc-2(e55) X; unc-64(sks4) III | UHN30 |
| unc-2(hp647) X; unc-64(sks4) III | UHN31 |
| unc-13(s69) I; unc-64(sks4) III | UHN32 |
| unc-13(e51) I; unc-64(sks4) III | UHN33 |
| unc-13(e1091) I; unc-64(sks4) III | UHN34 |
| unc-13(n2813) I; unc-64(sks4) III | UHN35 |
| tom-1(ok285) I; unc-64(sks4) III | UHN36 |
| unc-13(s69) tom-1(ok285) I; unc-64(sks4) III | UHN37 |
| unc-13(s69) tom-1(ok285) I; unc-18(sks2) X | UHN26 |
| unc-10(md1117) X; unc-64(sks4) III | UHN38 |
| unc-31(e928) IV; unc-64(sks4) III | UHN39 |
| snt-1(md290) II; unc-64(sks4) III | UHN40 |
| unc-18(md299) X; unc-64(sks4) III | UHN41 |
| unc-18(e81) X; unc-64(sks4) III | UHN42 |
| unc-18(sks1) X; unc-64(sks4) III | UHN43 |
| unc-49(e407) unc-64(sks4) III | UHN44 |
| unc-64(sks4) III; zxIs6 V | UHN45 |
| snt-1(md290) II; zxIs6 V | UHN46 |
| snt-1(md290) II; unc-64(sks4) III; zxIs6 V | UHN47 |
| unc-13(s69) I; zxIs6 V | UHN48 |
| unc-13(s69) I; unc-64(sks4) III; zxIs6 V | UHN49 |
| PCR primers used to verify unc-64(sks4) LE open mutant allele | |
| Primer name | Primer sequence |
| SS1727 | 5′-GGTGTAAGGGACGAATTCAGAG-3′ |
| SS1728 | 5′-CAAACCTGTTGGCTATCTGTGA-3′s |

**Aldicarb assays**. Aldicarb sensitivity was assessed using synchronously grown adult worms placed on non-seeded 30 mm NGM plates containing 0.3 mM or 1 mM aldicarb. All assays were done in 1 mM aldicarb plates unless specified otherwise. Over a 4 or 24 h period, worms were monitored for paralysis at 15 or 30 min intervals. Worms were considered paralyzed when there was no movement or pharyngeal pumping in response to three taps to the head and tail with a platinum wire. Once paralyzed, worms were removed from the plate. Six to 7 sets of 10–20 worms were examined for each strain.

**Growth speed analyses**. Growth speed was assessed by eye using an OMAX A3580U camera on a dissecting microscope. Briefly, 15 eggs were extracted from adult worms into OP50-seeded NGM plates and placed along the edge of the bacterial lawn roughly equidistant from each other. Photos of the plates were taken daily at ×1 magnification with the OMAX ToupView v3.7 software for up to 7 days for N2 worms, and up to 9 days for the other strains. Growth speed and body size were compared by eye based on photos taken on day 9.

**Electrophysiology**. The dissection of the *C. elegans* was described previously (Gao and Zhen[83], and Richmond et al.[12]). Briefly, 1 or 2 days old hermaphrodite adults were glued to a sylgard (Dow Corning, USA)-coated cover glass covered with bath solution. The integrity of the neuromuscular junction preparation was visually examined via DIC microscopy, and anterior muscle cells were patched using fire-polished 4–6 MΩ resistant borosilicate pipettes (World Precision Instruments, USA). Membrane currents were recorded in the whole-cell configuration by HEKA EPC-9 patch clamp amplifier, using the PULSE software (Version 8.74) and processed with Igor Pro 6.21 (WaveMetrics) and Clampfit 10.2 (Axon Instruments, Molecular Devices, USA). Data were digitized at 10 kHz and filtered at 2.6 kHz.

Light stimulation of *zxIs6* was performed with an light-emitting diode lamp at a wavelength of $460 \pm 5$ nm ($3.75$ mW/mm$^2$), triggered by the PULSE software for 10 ms. Muscle cells were recorded with a holding potential of $-60$ mV. The recording solutions were as described in our previous studies[83]. Specifically, the pipette solution contains (in mM): K-gluconate 115, KCl 25, CaCl$_2$ 0.1, MgCl$_2$ 5, BAPTA 1, HEPES 10, Na$_2$ATP 5, Na$_2$GTP 0.5, cAMP 0.5, cGMP 0.5 pH 7.2 with KOH, ~320 mOsm. The bath solution consists of (in mM): NaCl 150, KCl 5, CaCl$_2$ 5, MgCl$_2$ 1, glucose 10, sucrose 5, HEPES 15 pH 7.3 with NaOH, ~330 mOsm. All chemicals were from Sigma. Experiments were performed at room temperatures (20–22 °C).

**Protein expression and purification**. Bacterial expression and purification of full-length rat syntaxin-1A, the rat syntaxin LE mutant, a cysteine-free variant of full-length rat SNAP-25a, a full-length rat synaptobrevin-2, full-length rat Munc18-1, rat synaptotagmin-1 C$_2$AB fragment (residues 131–421), full-length rat Munc13-1 Chinese hamster NSF, full-length Bos Taurus αSNAP, and a rat Munc13-1 C$_1$C$_2$BMUNC$_2$C fragment (residues 529–1725, Δ1408–1452) were described previously (ref. [84] and references therein). We note that full-length rat syntaxin-1A was purified in buffer containing dodecylphosphocholine to prevent its aggregation[85].

**Lipid and content mixing assays**. Assays that simultaneously measure lipid and content mixing were performed as previously described[52]. Briefly, vesicle (V)-liposomes with reconstituted rat synaptobrevin-2 (protein-to-lipid ratio, 1:500) contained 39% palmitoyl-2-oleoyl-sn-glycero-3-phosphocholine (POPC), 19% 1,2-dioleoyl-sn-glycero-3-phospho-L-serine (DOPS), 19% palmitoyl-2-oleoyl-sn-glycero-3-phosphoethanolamine (POPE), 20% cholesterol, 1.5% 1,2-dipalmitoyl-sn-glycero-3-phosphoethanolamine-N-(7-nitro-2-1,3-benzoxadiazol-4-yl) (NBD-PE), and 1.5% Marina Blue DHPE. T-liposomes with reconstituted rat syntaxin-1A or open syntaxin-1A LE mutant together with rat SNAP-25A (protein-to-lipid ratio, 1:800) contained 38% POPC, 18% DOPS, 20% POPE, 20% cholesterol, 2% L-α-phosphatidylinositol-4,5-bisphosphate, and 2% palmitoyl-2-oleoyl-sn-glycerol. Lipid solutions were then mixed with respective proteins and with 4 µM Phycoerythrin-Biotin for T-liposomes or with 8 µM Cy5-Streptavidin for V-liposomes in 25 mM HEPES pH 7.4, 150 mM KCl, 1 mM tris(2-carboxyethyl) phosphine, 10% glycerol (v/v). V-liposomes (0.125 mM lipids) were mixed with T-liposomes (0.25 mM lipids) in a total volume of 200 µL in the presence of 2.5 mM MgCl$_2$, 2 mM ATP, 0.1 mM EGTA, 5 µM streptavidin, 0.4 µM NSF, 2 µM α-SNAP, 1 µM Munc18-1, 1 µM synaptotagmin-1 C$_2$AB, and 1 µM excess SNAP-25 with or without 0.2 µM Munc13-1 C$_1$C$_2$BMUNC$_2$C. Before mixing, T-liposomes were incubated with NSF, MgCl$_2$, ATP, EGTA, streptavidin, NSF, αSNAP, and Munc18-1 at 37 °C for 25 min. CaCl$_2$ (0.6 mM) was added at 300 s to each reaction mixture. A spectrofluorometer (Photon Technology International) was used to measure lipid mixing from de-quenching of the fluorescence of Marina Blue-labeled lipids (excitation at 370 nm, emission at 465 nm) and content mixing from the development of Förster resonance energy transfer between PhycoE-Biotin trapped in the T-liposomes and Cy5-streptavidin trapped in the V-liposomes (PhycoE-biotin excitation at 565 nm, Cy5-streptavidin emission at 670 nm). All experiments were performed at 30 °C. Lipid and content mixing were normalized as the percentage values of the maximum signals obtained by addition of 1% b-OG at the end to each reaction mixture (for lipid mixing) or to controls without streptavidin to measure maximal Cy5 fluorescence (for content mixing).

**Statistical analyses**. Statistical analyses were done in OriginPro2018 using the independent t-test for two-group experiments with a *p*-value < 0.05 as the threshold for statistical significance. For comparison of multiple groups, one-way analysis of variance was conducted, followed by Tukey's range test, with a significance level of 0.05.

**Reporting summary**. Further information on research design is available in the Nature Research Reporting Summary linked to this article.

## Data availability
The authors declare that data supporting the findings of this study are available within the paper and its Supplementary Information files. All additional information and the *C. elegans* mutants generated in this study will be made available upon reasonable request to the corresponding author. Source data are provided with this paper.

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

## Acknowledgements

This research was supported by the Natural Sciences and Engineering Research Council of Canada (RGPIN-2015-06438, to S.S.), the Canadian Institute of Health Research (MOP-130573 to S.S.), the Welch Foundation (I-1304 to J.R.) and the NIH (R35 NS097333 to J.R.), the National Natural Science Foundation of China (31871069 and 31671052 to S.G.), Major International (Regional) Joint Research Project (32020103007 to S.G.), the Junior Thousand Talents Program of China and funds from Huazhong University of Science and Technology (Dengfeng Initiative, Global Talents Recruitment Program). *unc-10(md1117)*, *unc-13(e51)*, *(e1091)*, *(n2813)*, *unc-31(e928)*, *unc-2(e55)*, *unc-18(md299) (e81)*, and *snt-1(md290)* worms were provided from the Caenorhabditis Genetics Center (University of Minnesota). VC223 strain with a genotype of *tom-1(ok285)* was provided by the *C. elegans* Reverse Genetics Core Facility at the University of British Columbia, which is part of the international *C. elegans* Gene Knockout Consortium. pTX21 plasmid and anti-syntaxin-1 poly-clonal antibody are kind gifts from Dr. Michael Nonet (Washington University) and Dr. Thomas C. Südhof (Stanford University), respectively. We also thank Dr. Thomas Südhof for comments on an earlier version of the draft. The summer scholarship for the Under-graduate Research Opportunity Program (UROP) from University of Toronto was awarded to C.-W.T. in 2016 and 2017.

## Author contributions

S.S., S.G., and J.R. designed and supervised experiments, analyzed data, and wrote the manuscript. C.-W.T., B.Y., M.H., and K.P.S. performed experiments, analyzed data, prepared figures, and wrote the manuscript. K.S. and X.X. contributed to the experiments. M.Z. provided the critical expertise and advice for the experiments, and helped with editing the manuscript. P.P.M. and L.H. helped with editing the manuscript.

## Competing interests

The authors declare no competing interests.
