## [Peer Review File · Nature Communications]

Reviewers' Comments:

Reviewer #1:

Remarks to the Author:

In this manuscript, the authors report about the rescue of different *C.elegans* mutants using expression of various mutants of the SNARE syntaxin I that stabilize the so-called "open" conformation. To this end, the mutant proteins were introduced in the *unc-64* endogenous locus using CRISPR/CAS9-directed homologous recombination, thus avoiding overexpression-related perturbations. In general, such mutations have been previously studied and thus the novelty of the findings may be challenged. However, there are too many controversies in the field, and this paper is of high quality and, at least in my opinion, clarifies some of the previous contradictions, and thus deserves publication after appropriate revision.

- In lines 122-123 and figure S1 c, the authors state that "open syntaxin/UNC-64 was expressed at slightly lower levels compared to wild-type..". This is not obvious from the figure. I suggest some kind of quantification, in particular for the *unc-64(sks4)* mutant.

- In line 183-186, the authors declare : " The key evidence that supported a specific genetic interaction between open syntaxin and UNC-13 derived from the observation that open syntaxin did not rescue exocytosis of the *unc-64(js21)* null mutant better than the wild-type syntaxin and did not overcome the impaired synaptic exocytosis of a severe loss-of-function mutant (e55) of *unc-2* (Richmond et al., 2001)". Please check. Richmond et al. showed that open syntaxin did not rescue synaptic exocytosis in the *unc-2* mutant, which suggests that it functions through the normal synaptic machinery.

- The authors presented evoked experiments for *snt-1* mutant. It would be interesting to present similar experiments for the other mutants, especially *unc-13*, that possesses C2 domains.

- In lines 287-288 and fig. 6f, the authors state that "...results suggest that open syntaxin and *tom-1* null exhibit additive, not synergistic effects on the rescue of *unc-13(s69)*.". If we take into account figure 5 b and figure 6 d, I disagree with this interpretation: They do look synergistic. Please explain

- The statement in line 359-362 has to be explained. I do not consider it necessary.

- Figure S9 should be included in the main text.

- In the liposome fusion assay, please show the decrease in the donor (PhycoE-biotin). Can you exclude that the increase in acceptor is due to light scattering by aggregation of liposomes?

- In line 1032-1033, change mM for μ M.

Reviewer #2:

Remarks to the Author:

In this manuscript, the authors created a knock-in strain of *C. elegans* with so-called "open" syntaxin-1 mutations (double mutations in the linker region). They went on to characterize the effects of the double mutations on synaptic release in WT and a variety of mutant worms. The majority of the findings, however, are not novel. In particular, the effects of the syntaxin-1 mutations on synaptic release, their ability to partially rescue *Unc13* mutations, and inability to rescue *Unc18* mutations, are well documented in literature. The ability of the syntaxin-1 mutations to partially rescue *syt-1* and calcium channel mutations is difficult to interpret and the authors did not provide a molecular explanation. Although the double mutations of syntaxin-1 are called open mutations, they likely affect other aspects of the protein. Moreover, the increased synaptic releases in the *syt-1* and calcium channel mutants may simply be caused by the double syntaxin-1 mutations, rather than any direct link between the mutations and *syt-1*/calcium channel. Overall, this reviewer feels that the manuscript lacks significant molecular insights and thus is more suitable for a specialized journal.

Reviewer #3:

Remarks to the Author:

This is a remarkable and thorough study characterizing the effects of a mutated open form of the SNARE protein syntaxin in synaptic exocytosis which is a key element in the current molecular models of neurotransmitter release. The authors have used CRISPR-mediated genome editing of *unc-64* (syntaxin) to generate knock-in worms expressing the open form of syntaxin. The functional characterization of a particularly wide spectrum of *C. elegans* mutants is based on (1) behavioral/neurological phenotypes: thrashing ability and paralysis in response to aldicarb and (2) spontaneous neurotransmitter release (only in one case they have analyzed evoked release). Liposome fusion assays provide robustness to the in vivo observations.

The authors report that the open syntaxin KI mutant exhibits a gain-of-function in synaptic exocytosis. Furthermore, the open syntaxin KI mutant rescues the synaptic and motor phenotype in a number of neurotransmitter release deficient mutants such as *snt-1/synaptotagmin*, *unc-2/P/Q/N* Ca²⁺ channel alpha-subunit, *unc-31/CAPS* or *tom-1/tomosyn*. The rescue is much weaker in other mutants such as *unc-10/RIM* and *unc-13*; indeed in most of *unc-13* mutants (*s69*, *e51* and *e1091*) there is basically no rescue. The open form of syntaxin aggravates the phenotype of *unc-18* mutants. The poor or non-existing rescue of *unc-13* mutants is a key point in this study because apparently contrasts with a previous seminal study (Richmond et al., Nature 2001) proposing that an open form of syntaxin bypasses the requirement for UNC-13 in vesicle priming. Other differences between the two studies include the gain-of-function of the syntaxin open form and the rescue of the *unc-2* mutants that are observed by Tien et al. but not by Richmond et al.

The apparent differences between the current study by Tien et al. and Richmond et al. are most likely caused by the different methodology used to generate the open form of syntaxin mutants: CRISPR-mediated genome editing (Tien) vs. multicopy expression of open syntaxin (Richmond). The results reported now by Tien et al. are certainly very relevant within the context of the molecular mechanisms of exocytosis and clearly important in the field. The study by Tien et al. is a mandatory study to further investigate the role of open syntaxin, initially proposed by Richmond et al. almost 20 years ago, but using now the state-of-the-art genome editing techniques. In this sense, both studies are certainly useful and informative to advance into the mechanisms of SNARE complex assembly. In any case, importantly, the results reported now by Tien et. support a key essential role in membrane fusion for UNC-13 beyond its proposed role to open syntaxin.

The conclusions are very well and elegantly discussed and supported by the convincing results. The paper is comprehensive and nicely written.

Comments

1. It is surprising the almost full rescue of thrashing activity observed in open syntaxin; *unc-2* (*e55*) mutants. I understand thrashing is a controlled movement that requires a neuronal command that ultimately would open Ca²⁺ channels before Ca²⁺-dependent exocytosis occurs. I would not expect this degree of rescue just by a mere increase of uncontrolled release downstream Ca²⁺-channels. In contrast to open syntaxin; *unc-2* (*e55*) mutants, the open syntaxin; *snt-1* mutants present only a rather moderate rescue of thrashing rates. How do the authors explain these results?

To address the major concerns raised by the reviewers, we have completed two key experiments and analyses. Specifically, (i) we performed several Western blot analyses and quantified the expression level of open syntaxin (=the UNC-64(sks4) mutant) in comparison with WT syntaxin (= WT UNC-64), which is shown in new Supplementary Fig. S1c, d and (ii) we measured evoked exocytosis of *unc-13(s69); open syntaxin KI* double mutant in comparison with *unc-13(s69)* (shown in Fig. 5k, l).

We also highlighted all changes in yellow in the manuscript text file and prepared the source file.

We believe that our manuscript is tremendously improved by fully addressing the referees' concerns and we hope that it is now acceptable to *Nature Communications*. Below is the point-by-point response to all the concerns raised by the reviews.

Sincerely,

Josep Rizo, Ph.D
Shangbang Gao, Ph.D
Shuzo Sugita, Ph.D

Reviewer #1:

In this manuscript, the authors report about the rescue of different *C.elegans* mutants using expression of various mutants of the SNARE syntaxin I that stabilize the so-called “open” conformation. To this end, the mutant proteins were introduced in the *unc-64* endogenous locus using CRISPR/CAS9-directed homologous recombination, thus avoiding overexpression-related perturbations. In general, such mutations have been previously studied and thus the novelty of the findings may be challenged. However, there are too many controversies in the field, and this paper is of high quality and, at least in my opinion, clarifies some of the previous contradictions, and thus deserves publication after appropriate revision.

-We appreciate the positive comments of this reviewer.

In lines 122-123 and figure S1 c, the authors state that “open syntaxin/UNC-64 was expressed at slightly lower levels compared to wild-type.” This is not obvious from the figure. I suggest some kind of quantification, in particular for the *unc-64(sks4)* mutant.

-We appreciate this constructive suggestion. We performed several (n=7) Western blot analyses and quantified the expression level of open syntaxin (=the UNC-64(sks4) mutant) in comparison with WT syntaxin (= WT UNC-64). Here, we used β -tubulin signals as loading controls. The quantification of the WT vs open syntaxin signals normalized by β -tubulin revealed that there was no significant difference between the open mutant and WT levels. These data are shown in Supplementary Fig. S1c, d. Interestingly, we observed that the signal of open syntaxin was present not only at the 35kDa monomer position but also at the ~80 kDa SNARE complex position despite that the worm sample was boiled once in SDS buffer. The signal at ~80 kDa disappeared after

repeated boiling of the samples. These results suggest that open syntaxin accelerates SNARE complex assembly as previously suggested by Garber et al. (2008, Science), which may be the reason why open syntaxin increases spontaneous and evoked release. We described these results on page 6 of the text.

In line 183-186, the authors declare : “ The key evidence that supported a specific genetic interaction between open syntaxin and UNC-13 derived from the observation that open syntaxin did not rescue exocytosis of the unc-64(js21) null mutant better than the wild-type syntaxin and did not overcome the impaired synaptic exocytosis of a severe loss-of-function mutant (e55) of unc-2 (Richmond et al., 2001)”. Please check. Richmond et al. showed that open syntaxin did not rescue synaptic exocytosis in the unc-2 mutant, which suggests that it functions through the normal synaptic machinery.

-Thank you for this comment. We realize that we used poor wording in this sentence; we did not mean to suggest that the unc-2 mutant suffers from an impaired exocytosis machinery. Therefore we have now corrected our writing in the following manner (from bottom of page 8 to top of page 9): "The key evidence that supported a specific genetic interaction between open syntaxin and UNC-13 derived from the observation that open syntaxin did not rescue exocytosis of the unc-64(js21) null mutant better than the wild-type syntaxin and **did not rescue the reduced exocytosis caused by reduced Ca^{2+} influx in a severe loss-of-function mutant (e55) of UNC-2 Ca^{2+} channels (Richmond et al., 2001).**"

The authors presented evoked experiments for snt-1 mutant. It would be interesting to present similar experiments for the other mutants, especially unc-13, that possesses C2 domains.

-We appreciate this constructive suggestion. To this end, we generated two new lines, namely the unc-13 mutant and the unc-13; open syntaxin KI double mutant which express channel rhodopsin in cholinergic neurons, and we measured light evoked neurotransmitter release from these worms. We found that, while there was almost no evoked release in unc-13 mutant, open syntaxin significantly increased evoked release in the unc-13 background. Yet the evoked release of the unc-13; open syntaxin double mutant is <10% of that of N2 WT and comparable with that of the open syntaxin; snt-1 double mutant. These new data are included as Fig. 5k, l and described on page 11 of the text.

In lines 287-288 and fig. 6f, the authors state that “...results suggest that open syntaxin and tom-1 null exhibit additive, not synergistic effects on the rescue of unc-13(s69).”. If we take into account figure 5 b and figure 6 d, I disagree with this interpretation: They do look synergistic. Please explain

-We agree with the reviewer's comments and have revised the text accordingly. We now state “open syntaxin and tom-1 null exhibit a synergistic effect yet this synergism is weaker than that of unfurled unc-18 and tom-1 null.” (page 13)

The statement in line 359-362 has to be explained. I do not consider it necessary. -As suggested, we removed the statement in lines 359-362 of our original text.

Figure S9 should be included in the main text.

-In accordance with the reviewer's suggestion, we have shown old Figure S9 as new Figure 9.

In the liposome fusion assay, please show the decrease in the donor (PhycoE-biotin). Can you exclude that the increase in acceptor is due to light scattering by aggregation of liposomes?

*-The reconstitution assays that we perform are based on the method described in Zucchi and Zick [Mol Biol Cell 22, 4635-4646 (2011)] to monitor lipid mixing and content mixing simultaneously. The method relies on the use of two donor-acceptor FRET pairs (Marina Blue-NBD and PhycoE-Cy5), monitoring lipid mixing from the decrease in the fluorescence intensity of the Marina Blue donor that has the shorter wavelength (465 nm) and content mixing from the increase in the fluorescence intensity of the Cy5 acceptor that has the longer wavelength (670 nm). This approach avoids interference between the signals reporting lipid and content mixing, and has been widely used by Bill Wickner's lab at Dartmouth to study yeast vacuolar fusion and by Josep Rizo's lab to study synaptic vesicle fusion. Analysis of liposome aggregation under the conditions of the fusion assays demonstrated that the lipid and content mixing signals cannot arise from such aggregation [Liu et al. *elife* 5, e13696 (2016)]. Note in particular that the aggregation was induced by Munc13-1 and occurred to comparable extents in the absence and presence of Ca^{2+} , but no content mixing signal was observed before addition of Ca^{2+} in experiments performed with WT syntaxin-1, Munc18-1 and Munc13-1, or in controls that lacked Munc18-1 before and after addition of Ca^{2+} (Fig. 8b).*

In line 1032-1033, change mM for μ M.

-We changed mM for μ M.

Reviewer #2:

In this manuscript, the authors created a knock-in strain of *C. elegans* with so-called "open" syntaxin-1 mutations (double mutations in the linker region). They went on to characterize the effects of the double mutations on synaptic release in WT and a variety of mutant worms. The majority of the findings, however, are not novel. In particular, the effects of the syntaxin-1 mutations on synaptic release, their ability to partially rescue Unc13 mutations, and inability to rescue Unc18 mutations, are well documented in literature. The ability of the syntaxin-1 mutations to partially rescue syt-1 and calcium channel mutations is difficult to interpret and the authors did not provide a molecular explanation. Although the double mutations of syntaxin-1 are called open mutations, they likely affect other aspects of the protein. Moreover, the increased synaptic releases in the syt-1 and calcium channel mutants may simply be caused by the double syntaxin-1 mutations, rather than any direct link between the mutations and syt-1/calcium channel. Overall, this reviewer feels that the manuscript lacks significant molecular insights and thus is more suitable for a specialized journal.

- We respectfully disagree with this assessment of our paper. It is true that the observations that open syntaxin-1 can partially rescue unc-13 phenotypes and not unc-18 phenotypes were already described in the literature, but these findings led to the belief that open syntaxin-1 had a specific genetic interaction with unc-13. Our paper now

*shows that open syntaxin-1 can rescue the phenotypes arising from mutations in diverse proteins, including not only unc-13 but also snt-1 (synaptotagmin-1) and a unc-2 (N-type Ca²⁺ channel) mutants. Further, whereas previous studies did not see a difference in exocytosis between N2 WT vs the overexpressed open syntaxin rescued worms (Richmond et al., 2001, Nature), we saw significant increases of exocytosis (both spontaneous and evoked) by open syntaxin knockin (KI) compared with N2 WT. Thus, while previous studies emphasized the specific genetic interaction between open syntaxin-1 and unc-13 mutant, our results suggest instead a specific interaction of open syntaxin-1 with unc-18 and show that open syntaxin-1 can enhance neurotransmitter release in a wide variety of genetic backgrounds. This is an important paradigm shift in our understanding of synaptic vesicle exocytosis. This importance is recognized by reviewer 3, who states that **'The results reported now by Tien et al. are certainly very relevant within the context of the molecular mechanisms of exocytosis and clearly important in the field.'***

We also would like to emphasize that our overall results have a natural molecular explanation because SNARE complexes are known to be critical for neurotransmitter release and the open syntaxin-1 mutation is known to facilitate SNARE complex formation (Garber et al., 2008, Science; Also see our Supplementary Fig. S1c). Hence, the increased number of SNARE complexes resulting from the mutation can readily explain an increase in neurotransmitter release under diverse genetic backgrounds. The lack of rescue in unc-18 phenotypes is also readily explained because unc-18-1 plays a fundamental role in organizing SNARE complex formation that cannot be bypassed by helping to open syntaxin-1.

Nevertheless, we agree with the reviewer that we did not discuss the potential mechanisms underlying the rescue of synaptotagmin-1 (snt-1) and calcium channel (unc-2) phenotypes by open syntaxin sufficiently. Regarding the rescue of unc-2, Reviewer #3 also asked us the molecular mechanism why open syntaxin can rescue the exocytosis phenotype of the unc-2(e55) null mutant.

*We suggest that there is some residual calcium influx in the unc-2(e55) null mutant via other Ca²⁺ channels, namely EGL-19, and propose that the open syntaxin-1 mutant overcomes the reduced Ca²⁺ influx in unc-2(e55) mutant by increasing the Ca²⁺ sensitivity of the release machinery, which was actually shown in Gerber et al. (2008, Science). There are at least two additional Ca²⁺ channels (EGL-19, L-type; CCA-1, T-type) in addition to UNC-2 (N-type) in the genome of *C. elegans* (Bargmann et al., 1998, Science). Importantly, Liu et al. (2014, Nature Commun.) showed that SLO-2 K⁺ channels regulate neurotransmitter release from cholinergic and GABAergic motor neurons and that the activity of SLO-2 in motor neurons depends on Ca²⁺ entry through EGL-19, but not other Ca²⁺ channels (CCA-1 or UNC-2). Thus, EGL-19 Ca²⁺ channels play a regulatory role in neurotransmitter release from motor neurons, although the direct evidence of EGL-19 in neurotransmitter release has been difficult to show due to the lethality of egl-19 null mutant (Lee et al., 1997, EMBO J). Therefore, while UNC-2 is the major Ca²⁺ channel that controls neurotransmitter release, EGL-19 seems to play an additional role. This also explains why null mutant of unc-2 (=unc-2(e55)) still shows significant residual acetylcholine release and motility (see our Fig. 2). We further suggest that open syntaxin*

overcomes the lack of UNC-2 by increasing the sensitivity of the release machinery to residual Ca²⁺ influx from EGL-19 channels. Similarly, there are several synaptotagmin isoforms in the genome of C. elegans. These isoforms may replace SNT-1 and function similarly but not very efficiently, and an increased number of SNARE complexes can compensate for this defect. We included these discussions on pages 21-22.

We wonder what the basis is for the statement made by the reviewer that the double mutations of syntaxin-1 likely affect other aspects of the protein and for the implication that these unspecified effects underlie the increased synaptic release in the synaptotagmin-1 and calcium channel mutants. It would have been helpful if the reviewer would have provided examples of what aspects he/she was referring to, as it is impossible for us to address a criticism stated in such abstract terms. We do want to point out that there have been many studies of the syntaxin-1 open (LE) mutant for the past twenty years and we are not aware of any report suggesting that the LE mutation causes unexpected biochemical behaviors that cannot be explained by the well-characterized tendency of this mutation to open syntaxin-1.

Reviewer #3:

This is a remarkable and thorough study characterizing the effects of a mutated open form of the SNARE protein syntaxin in synaptic exocytosis which is a key element in the current molecular models of neurotransmitter release. The authors have used CRISPR-mediated genome editing of unc-64 (syntaxin) to generate knock-in worms expressing the open form of syntaxin. The functional characterization of a particularly wide spectrum of C. elegans mutants is based on (1) behavioral/neurological phenotypes: thrashing ability and paralysis in response to aldicarb and (2) spontaneous neurotransmitter release (only in one case they have analyzed evoked release). Liposome fusion assays provide robustness to the in vivo observations.

The authors report that the open syntaxin KI mutant exhibits a gain-of-function in synaptic exocytosis. Furthermore, the open syntaxin KI mutant rescues the synaptic and motor phenotype in a number of neurotransmitter release deficient mutants such as snt-1/synaptotagmin, unc-2/P/Q/N Ca²⁺ channel alpha-subunit, unc-31/CAPS or tom-1/tomosyn. The rescue is much weaker in other mutants such as unc-10/RIM and unc-13; indeed in most of unc-13 mutants (s69, e51 and e1091) there is basically no rescue. The open form of syntaxin aggravates the phenotype of unc-18 mutants. The poor or non-existing rescue of unc-13 mutants is a key point in this study because apparently contrasts with a previous seminal study (Richmond et al., Nature 2001) proposing that an open form of syntaxin bypasses the requirement for UNC-13 in vesicle priming. Other differences between the two studies include the gain-of-function of the syntaxin open form and the rescue of the unc-2 mutants that are observed by Tien et al. but not by Richmond et al.

The apparent differences between the current study by Tien et al. and Richmond et al. are most likely caused by the different methodology used to generate the open form of

syntaxin mutants: CRISPR-mediated genome editing (Tien) vs. multicopy expression of open syntaxin (Richmond). The results reported now by Tien et al. are certainly very relevant within the context of the molecular mechanisms of exocytosis and clearly important in the field. The study by Tien et al. is a mandatory study to further investigate the role of open syntaxin, initially proposed by Richmond et al. almost 20 years ago, but using now the state-of-the-art genome editing techniques. In this sense, both studies are certainly useful and informative to advance into the mechanisms of SNARE complex assembly. In any case, importantly, the results reported now by Tien et. support a key essential role in membrane fusion for UNC-13 beyond its proposed role to open syntaxin.

The conclusions are very well and elegantly discussed and supported by the convincing results. The paper is comprehensive and nicely written.

-We appreciate very much the enthusiastic comments by this reviewer.

Comments

1. It is surprising the almost full rescue of thrashing activity observed in open syntaxin; *unc-2* (e55) mutants. I understand thrashing is a controlled movement that requires a neuronal command that ultimately would open Ca²⁺ channels before Ca²⁺-dependent exocytosis occurs. I would not expect this degree of rescue just by a mere increase of uncontrolled release downstream Ca²⁺-channels. In contrast to open syntaxin; *unc-2* (e55) mutants, the open syntaxin; *snt-1* mutants present only a rather moderate rescue of thrashing rates. How do the authors explain these results?

*-This is an important point to consider. We have now measured evoked release of the *unc-13*; open syntaxin KI double mutant in response to the comments of reviewer #1. We found that *unc-13*; open syntaxin KI significantly increased evoked release compared with the *unc-13* single mutant, yet the evoked release of the *unc-13*; open syntaxin mutant is less than ~10% of the WT, which is similar to the exocytosis of the *snt-1*; open syntaxin double mutant. Then question arises as to why open syntaxin can almost fully rescue *unc-2*(e55) null mutant, as mentioned by the reviewer.*

*As explained above in our responses to the concerns from Reviewer 2, we suggest that there is still some calcium influx in the *unc-2*(e55) null mutant via the other Ca²⁺ channels, namely EGL-19, and propose that the open syntaxin-1 mutant overcomes the reduced Ca²⁺ influx in *unc-2*(e55)mutant by increasing the Ca²⁺ sensitivity of the release machinery, which was actually shown in Gerber et al. (2008, Science). There are at least two additional Ca²⁺ channels (EGL-19, L-type; CCA-1, T-type) in addition to UNC-2 (N-type) in the genome of *C. elegans* (Bargmann et al., 1998, Science). Importantly, Liu et al. (2014, Nature Commun) showed that SLO-2 K⁺ channels regulate neurotransmitter release from cholinergic and GABAergic motor neurons and that the activity of SLO-2 in motor neurons depends on Ca²⁺ entry through EGL-19, but not other Ca²⁺ channels (CCA-1 or UNC-2). Thus, EGL-19 Ca²⁺ channels play a regulatory role in neurotransmitter release from motor neurons, although the direct evidence of EGL-19 in neurotransmitter release had been difficult to show due to the lethality of *egl-19* null mutant (Lee et al., 1997, EMBO J). Therefore, while UNC-2 is the major Ca²⁺ channel that controls neurotransmitter release, EGL-19 seems to play an additional role. This also explains why null mutant of *unc-2* (=unc-2(e55)) still shows significant residual*

acetylcholine release and motility (see our Fig. 2). We further suggest that open syntaxin overcomes the lack of UNC-2 by increasing the sensitivity of the release machinery to residual Ca²⁺ influx from EGL-19 channels. We have included this discussion on page 21-22 in the discussion.

Reviewers' Comments:

Reviewer #1:

Remarks to the Author:

During revision the authors have addressed all points raised in my comments in full, partially by providing additional data. Therefore, I recommend acceptance of the manuscript.

Reviewer #2:

Remarks to the Author:

I do not question the experiments of this manuscript but I still have serious reservations about their interpretation of the data. The entire paper is based on presumably "open" syntaxin-1 mutations, which clearly affect many molecular interactions. Thus, it is difficult to determine how the mutations influence synaptic release. While I cannot recommend this manuscript for publication in Nature Communications, I can see one potential positive aspect of this work is to stimulate further studies in this important field.

Reviewer #3:

Remarks to the Author:

The authors have properly addressed my concerns and I do not have further comments.

Reviewer #1:

During revision the authors have addressed all points raised in my comments in full, partially by providing additional data. Therefore, I recommend acceptance of the manuscript.

-We are very grateful for the constructive and helpful comments of the reviewer, which helped to considerably improve the manuscript.

Reviewer #2

I do not question the experiments of this manuscript but I still have serious reservations about their interpretation of the data. The entire paper is based on presumably “open” syntaxin-1 mutations, which clearly affect many molecular interactions. Thus, it is difficult to determine how the mutations influence synaptic release. While I cannot recommend this manuscript for publication in Nature Communications, I can see one potential positive aspect of this work is to stimulate further studies in this important field.

-We appreciate that the reviewer now acknowledges the importance of our work.

The reviewer appears to suggest that the conformational effects of the LE mutation are uncertain and that the mutation can affect many interactions, which would cloud the interpretation of the data. However, using NMR spectroscopy, we unambiguously established a long time ago that WT syntaxin-1 adopts a closed conformation and that the LE mutation facilitates opening of this conformation (Dulubova et al., 1999, EMBO J). Opening the conformation of syntaxin-1 is predicted to facilitate SNARE complex assembly, which was later confirmed experimentally in Gerber et al. (2008, Science) and other subsequent papers. Moreover, the mutated residues are buried in closed syntaxin-1, as shown by the crystal structure of the syntaxin-1-Munc18-1 complex (Misura et al. 2000), and these residues have not been implicated in any interactions with known components of the release machinery. Thus, the structural and biochemical consequences of the LE mutation have been well-defined.

Reviewer #3

The authors have properly addressed my concerns and I do not have further comments.

-We are very grateful for the constructive and helpful comments of the reviewer, which helped to considerably improve the manuscript.